# M³PL: Identifying and Exploiting View Bias of Prompt Learning

**Chujie Zhao**[*]                    *zhaocj22@mails.tsinghua.edu.cn*
*Department of Automation*
*Tsinghua University, Beijing, China*

**Tianren Zhang**[*]                    *zhangtr22@mails.tsinghua.edu.cn*
*Department of Automation*
*Tsinghua University, Beijing, China*

**Guanyu Chen**                    *chen-gy23@mails.tsinghua.edu.cn*
*Department of Automation*
*Tsinghua University, Beijing, China*

**Yizhou Jiang**                    *jiangyz20@mails.tsinghua.edu.cn*
*Department of Automation*
*Tsinghua University, Beijing, China*

**Feng Chen**[†]                    *chenfeng@mail.tsinghua.edu.cn*
*Department of Automation*
*Tsinghua University, Beijing, China*

**Reviewed on OpenReview:** *https://openreview.net/forum?id=2rnTIBm19V*

## Abstract

Prompt learning is an effective means of fine-tuning multi-modal foundation models such as CLIP. Despite existing success, the inner mechanism of multi-modal prompt learning has not been well understood. In this work, we identify an inductive bias of multi-modal prompt learning, which we refer to as view bias, that the learned prompts may extract only a partial subset of useful features (views) and ignore others. This bias can undermine the model's generalization ability, particularly under distribution shifts. We further observe that independently trained prompts have distinct view biases, contrary to the existing belief that they may converge to similar local optima due to having the same cross-modal representation matching objective. Based on our observations, we propose **M**ulti-modal **M**atching **M**ulti-**P**rompt **L**earning (M³PL), which incorporates multiple paired prompts and a cross-modal contrastive regularizer that facilitates the prompt pairs to encapsulate a broader spectrum of views. Extensive experiments show that M³PL effectively boosts the model's generalization capability, achieving state-of-the-art performance under various distribution shifts.

## 1 Introduction

Recent advancements in Vision-Language pre-trained Models (VLMs) such as CLIP (Radford et al., 2021) and ALIGN (Jia et al., 2021) have demonstrated impressive open-vocabulary generalization capabilities across various downstream tasks (Li et al., 2022b; Ramesh et al., 2022; Tevet et al., 2022). However, the large scale of VLMs and the scarcity of high-quality training data often make fine-tuning the entire model costly. In response, prompt learning, which appends additional, learnable continuous vectors (prompts)

---

[*]Equal contribution
[†]Corresponding author

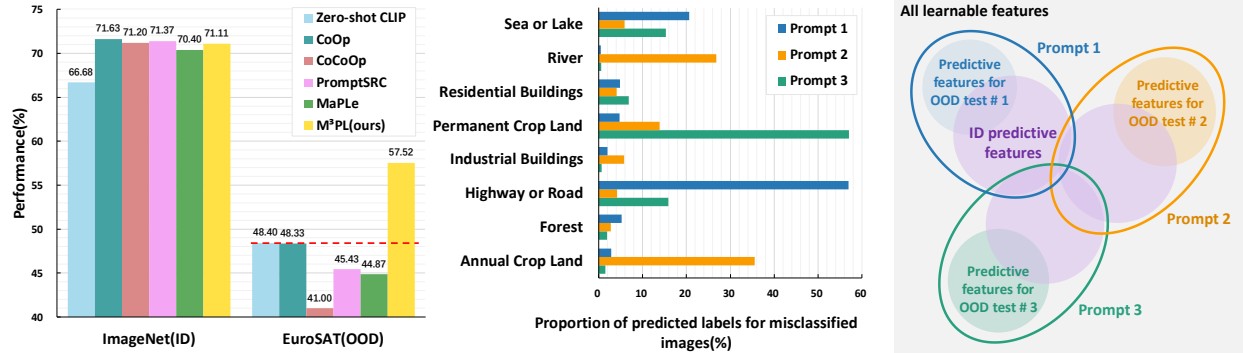

Figure 1: **(Left)** Zero-shot performance of CLIP ViT-B/16 on EuroSAT after few-shot fine-tuning on ImageNet. Existing prompt learning methods compromise the original generalization capability of CLIP. **(Middle)** The distribution of predicted labels for misclassified images on EuroSAT of three independently trained prompts with nearly identical ID test accuracies. Different prompts exhibit distinct predictive distributions. **(Right)** An illustration of view bias of different prompts. Each prompt only learns a partial set of OOD-useful features and thus can only solve certain OOD tasks but not all of them.

to VLMs while keeping pre-trained weights intact, has emerged as an efficient alternative for fine-tuning VLMs (Zhou et al., 2022b; Khattak et al., 2023b; Lu et al., 2022; Khattak et al., 2023a; Zhou et al., 2022a).

Although previous prompt learning methods (Zhou et al., 2022b;a) have significantly enhanced the in-distribution (ID) performance of the fine-tuned models, their improvements in out-of-distribution (OOD) settings are still limited. In particular, on datasets where both image and text exhibit substantial distribution shifts, existing prompt-based methods may even *underperform* zero-shot CLIP. As an example, in the EuroSAT (Helber et al., 2019) satellite dataset, existing methods reduce the OOD accuracy of CLIP by 0.07% to 7.40% after few-shot fine-tuning on ImageNet (Deng et al., 2009), as shown in Figure 1 (left).

Why do existing prompt learning methods reduce OOD robustness? To answer this question, it is necessary to analyze what is actually learned by prompts and how it relates to generalization. In existing work, it is believed that what is learned by prompts is roughly uniquely determined by the training data and the prompt learning objective (Chen et al., 2023). However, through an empirical study of the mistakes made by independently trained prompts in OOD settings, we challenge this belief. In particular, we observe that *prompts with nearly identical ID accuracies can make very different OOD mistakes*. For example, as illustrated in Figure 1 (middle), a set of learned prompts with almost the same ID performance exhibits distinct incorrect image predictions. This phenomenon implies that prompts optimized under the same conditions may converge to different local optima, where the model use *different features* for prediction. As will be detailed in Section 4, similar phenomena also manifest in many datasets with distribution shifts.

To investigate the inner mechanism of the above phenomenon, we need to first characterize the learned features of multi-modal prompts. However, the existing analysis is also limited in this direction: to our knowledge, the most relevant work is by Oymak et al. (2023), which studies uni-modal instead of multi-modal prompt learning. Moreover, they focus on analyzing the roles of the attention mechanism in prompt learning without characterizing the learned features of the prompts. To overcome this limitation, we theoretically analyze multi-modal prompt learning under a structured feature model. Compared to the work by Oymak et al. (2023), our analysis draws inspiration from recent studies on the feature learning process of neural networks (Allen-Zhu & Li, 2023; Shah et al., 2020) and analyzes the interaction between prompts and inputs in different modalities. Through our analysis, we show that (1) prompt learning can be viewed as a *feature selection* process that selects pre-trained features to match visual and textual representations on downstream tasks, and (2) due to the *multi-solution* nature of the feature selection schemes, prompts may only select a subset of useful features (views), which we term as *view bias*. For ID data, since the features useful for prediction are often redundant (Guyon & Elisseeff, 2003), view bias does not impact test performance much and may even mitigate overfitting. However, in OOD scenarios where not all features useful in ID data are

still predictive, view bias can lead to the lack of predictive features, thus limiting the generalization ability. This is consistent with our empirical observation that prompts with different OOD mistakes can still achieve similar ID accuracies. Please see Figure 1 (right) for an illustration.

Based on the analysis, we propose a principled **M**ulti-modal **M**atching **M**ulti-**P**rompt **L**earning (M$^3$PL) method to mitigate the adverse effect of view bias in OOD generalization. The main idea of M$^3$PL is to aggregate different views from multiple independently optimized multi-modal prompt pairs. Leveraging the fact that independently trained prompt pairs tend to have different view biases, M$^3$PL can obtain a diverse and rich collection of useful views through aggregation, hence improving generalization under distribution shifts. However, ensembling multiple prompt pairs may also lead to view redundancy that can harm generalization. To mitigate this problem, we further introduce a cross-modal contrastive regularizer to facilitate distinct views for different prompts, which also enhances the model's OOD robustness. Empirically, on the cross-dataset setting with distribution shifts in both visual and textual domains, M$^3$PL achieves a ***3.5x*** increase over previous methods in OOD accuracy gains over zero-shot CLIP, and significantly outperforms prior methods in terms of a complementary OOD performance measure named *effective robustness ratio*.

In summary, our main contributions are three-fold:

- We identify a failure mode in existing prompt learning methods under large distribution shifts caused by view bias, which provides new insights into analyzing the generalization of prompt learning.

- We theoretically analyze multi-modal attention-based prompt learning, which explains the view bias phenomenon and lays a foundation for future analysis.

- We propose a theoretically grounded and minimally constrained prompt learning framework, M$^3$PL, which achieves state-of-the-art performance in average test accuracy and effective robustness ratio across three common OOD generalization settings.

## 2    Related Work

**Vision-language pre-trained models and downstream task adaptation.**    Vision-Language pre-trained Models (VLMs) have achieved remarkable performance in few-shot and zero-shot recognition tasks by leveraging large-scale image-text paired training data to align vision and text representations (Li et al., 2021; Jia et al., 2021; Radford et al., 2021; Kim et al., 2021). Supported by the expressive power of language, VLMs gain an understanding of open-world visual concepts, enabling them to adapt to various applications, including object detection and segmentation (Li et al., 2022a; Xu et al., 2022; Gu et al., 2022; Li et al., 2022b), image generation (Ramesh et al., 2022; Patashnik et al., 2021), action recognition (Tevet et al., 2022; Wang et al., 2021), etc. While VLMs provide generalizable representations, how to efficiently adapt them to downstream tasks remains an important challenge. Prior work has proposed parameter-efficient tuning methods based on CLIP, including adapter-based (Gao et al., 2023; Zhang et al., 2022) and prompt-based methods (Zhou et al., 2022b;a). Our work introduces a multi-modal multi-prompt learning framework that, while maintaining parameter-efficiency during adaptation, enhances the robustness of the adapted models.

**Prompt learning.** Prompt learning originated in the NLP domain. Early methods used expert knowledge to manually construct prompts, also known as prompt engineering (Brown et al., 2020; Petroni et al., 2019). Later, Jiang et al. (2020); Shin et al. (2020) proposed to automatically search for templates, and Li & Liang (2021); Tsimpoukelli et al. (2021); Liu et al. (2023); Lester et al. (2021) extended the search to the continuous representation space. Recently, prompt learning has been introduced to vision tasks. Jia et al. (2022) incorporated learnable prompts in vision models. CoOp (Zhou et al., 2022b) and CoCoOp (Zhou et al., 2022a) add a learnable single prompt in the language branch of CLIP. MaPLe (Khattak et al., 2023a) extends this approach to both vision and language branches. PromptSRC (Khattak et al., 2023b) incorporates self-regularization into the prompt learning process. ProDA (Lu et al., 2022) and PLOT (Chen et al., 2023) learn multiple prompts only in the language branch; ProDA assumes a Gaussian distribution for prompts, while PLOT employs a two-stage optimization strategy based on local features and optimal transport theory. Unlike these methods, M$^3$PL does not require modifying the objective or assuming a parameter distribution, enabling the learning of diverse prompts in a simpler and minimally constrained manner. Wang et al. (2023)

attempts to utilize multiple soft text prompts and fine-tune a linear classifier on them, while our M$^3$PL adopts the CoOp paradigm of optimizing the prefix of the prompt. On the theoretical side, there is little work analyzing prompt learning, even outside the multi-modal setting. The recent work by Oymak et al. (2023) analyzes the role of attention in prompt learning. However, their analysis focuses on the single-modal setting and considers a simplified attention model where learnable tokens are only appended to queries but not keys and values, which deviates from the multi-modal prompt learning practice.

## 3  Preliminaries

This section briefly reviews the prompt learning framework based on CLIP, which our method is built upon. Empirically, we can also apply our method to other image-text pre-trained backbones such as SigLIP (Zhai et al., 2023) in a similar fashion. Here we only focus on CLIP for simplicity.

**CLIP architecture.** CLIP comprises both an image encoder and a text encoder and performs zero-shot classification by matching the visual representation with different textual representations corresponding to different labels. Our implementation is based on CLIP with Vision Transformer (ViT) (Dosovitskiy et al., 2021) as its image encoder. Concretely, denote CLIP's image encoder as $f$ and text encoder as $g$, with parameters denoted by $\boldsymbol{\theta}_f$ and $\boldsymbol{\theta}_g$, respectively. Both encoders consist of $L$ multi-head self-attention layers. In the vision branch, the input image $\boldsymbol{X}$ is initially divided into $N$ fixed-size patches $\{\boldsymbol{x}_1, \ldots, \boldsymbol{x}_N\}$. Next, this patch sequence is embedded as tokens $\{\boldsymbol{z}_1, \ldots, \boldsymbol{z}_N\}$ and concatenated with a learnable classification token $\boldsymbol{z}^0_{\text{cls}}$ to form the input sequence $\boldsymbol{Z}_0 = \{\boldsymbol{z}^0_{\text{cls}}, \boldsymbol{z}^0_1, \ldots, \boldsymbol{z}^0_N\}$ of the first multi-head self-attention layer. Similarly, we denote $\boldsymbol{Z}_i = \{\boldsymbol{z}^i_{\text{cls}}, \boldsymbol{z}^i_1, \ldots, \boldsymbol{z}^i_N\}$ as the input sequence for the $(i+1)$-th layer. Finally, the classification token $\boldsymbol{z}^L_{\text{cls}}$ from the output of the $L$-th transformer layer is mapped to a $d$-dimensional vector in CLIP's aligned representation space, serving as the visual representation $\boldsymbol{v} = f(\boldsymbol{Z}_0; \boldsymbol{\theta}_f) \in \mathbb{R}^d$. In the language branch, the label is concatenated with a fixed template, such as "a photo of {$label$}", to serve as input. This input is then tokenized and embedded to form the input text token sequence $\boldsymbol{W}_0 = \{\boldsymbol{w}^0_{\text{SOS}}, \boldsymbol{w}^0_1, \ldots, \boldsymbol{w}^0_K, \boldsymbol{w}^0_y, \boldsymbol{w}^0_{\text{EOS}}\}$ (assuming the template has $K$ tokens), where $\boldsymbol{w}^0_y$ represents the token corresponding to the class label. Similar to the vision side, the $\boldsymbol{w}^L_{\text{SOS}}$ from the output of the $L$-th transformer layer is mapped to a $d$-dimensional vector, serving as the textual representation $\boldsymbol{t} = g(\boldsymbol{W}_0; \boldsymbol{\theta}_g) \in \mathbb{R}^d$.

**CLIP for image classification.** For classification, assuming a set of $C$ candidate class labels $\{y_1, \ldots, y_C\}$, the probability of a CLIP-based model predicting the label as $y_k$ is then given by:

$$p(\hat{y} = y_k \mid \boldsymbol{X}) = \frac{\exp(\cos(\boldsymbol{v}, \boldsymbol{t}_k)/\tau)}{\sum_{i=1}^C \exp(\cos(\boldsymbol{v}, \boldsymbol{t}_i)/\tau)} \tag{1}$$

where $\cos(\cdot, \cdot)$ denotes the cosine similarity, $\tau$ is a temperature parameter, $\boldsymbol{v}$ is the output of the input image, and $\boldsymbol{t}_k$ represents the textual representation corresponding to label $y_k$.

**Prompt learning based on CLIP.** We employ the basic method of Independent Vision-Language Prompting (IVLP) (Rasheed et al., 2023) to elucidate the fundamental principles of prompt learning. At the input layer, $N_v$ and $N_t$ learnable tokens serve as visual and textual prompts, denoted by $\boldsymbol{p}^0_v$ and $\boldsymbol{p}^0_t$, respectively. In the vision branch, $\boldsymbol{p}^0_v$ is concatenated directly with $\boldsymbol{Z}_0$, while in the language branch, $\boldsymbol{p}^0_t$ replaces the corresponding tokens in $\boldsymbol{W}_0$, resulting in new input sequences $\widetilde{\boldsymbol{Z}}_0 = \{\boldsymbol{p}^0_v, \boldsymbol{Z}_0\}$ and $\widetilde{\boldsymbol{W}}_0 = \{\boldsymbol{w}^0_{\text{SOS}}, \boldsymbol{p}^0_t, \boldsymbol{w}^0_y, \boldsymbol{w}^0_{\text{EOS}}\}$. Given a prompt depth $J$, prompts will be added to the first $J$ layers of the transformer. At the $i$-th layer, the input sequences are $\widetilde{\boldsymbol{Z}}_{i-1} = \{\boldsymbol{p}^{i-1}_v, \boldsymbol{Z}_{i-1}\}$ and $\widetilde{\boldsymbol{W}}_{i-1} = \{\boldsymbol{w}^{i-1}_{\text{SOS}}, \boldsymbol{p}^{i-1}_t, \boldsymbol{w}^{i-1}_y, \boldsymbol{w}^{i-1}_{\text{EOS}}\}$. Note that the output tokens at the positions of the previous layer's prompts are replaced with new learnable tokens added in the subsequent layer. Ultimately, we obtain the visual and textual representations denoted by $\tilde{\boldsymbol{v}} = f(\widetilde{\boldsymbol{Z}}_0; \boldsymbol{\theta}_f, \{\boldsymbol{p}^i_v\}_{i=0}^{J-1})$ and $\tilde{\boldsymbol{t}} = g(\widetilde{\boldsymbol{W}}_0; \boldsymbol{\theta}_g, \{\boldsymbol{p}^i_t\}_{i=0}^{J-1})$, respectively. During training, the pre-trained parameters $\boldsymbol{\theta}_f$ and $\boldsymbol{\theta}_g$ are frozen and only the learnable prompts are optimized.

## 4  Empirical Evidence of View Bias

This section details the experimental settings and main observations in our empirical study in Section 1 and presents additional evidence of the view bias problem in more datasets.

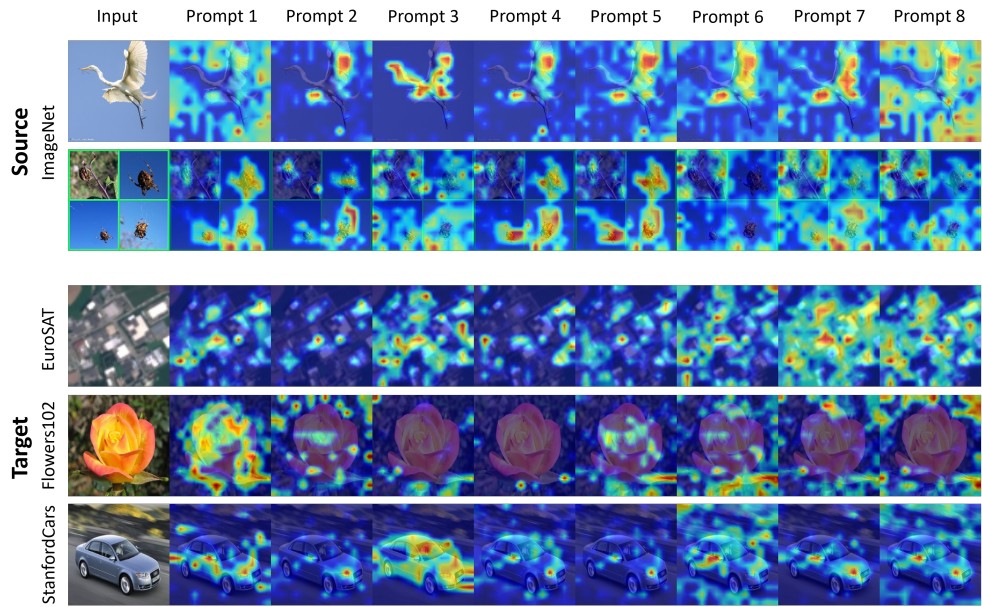

Figure 2: Visualization of attention heatmaps of different prompts on test images in ImageNet and three different target datasets. Different prompts focus on varying regions of the images, with significant differences in the attention distribution. All prompts are learned using identical training data.

Table 1: Relative average Jenson-Shannon (JS) divergence of the predicted label distributions of different prompts. A larger value indicates greater disparity. We also report the relative average JS divergence (compared to the JS divergence on the EuroSAT dataset without M³PL) between the aggregated prompts trained with different initializations using our proposed M³PL method.

| | Caltech 101 | Oxford Pets | Stanford Cars | Flowers 102 | Food101 | FGVC Aircraft | SUN397 | DTD | EuroSAT | UCF101 | *Average* |
|---|---|---|---|---|---|---|---|---|---|---|---|
| Average JS Divergence | 1.043 | 0.520 | 0.590 | 0.850 | 0.254 | 1.470 | 0.358 | 0.516 | 1.000 | 0.579 | 0.718 |
| + M³PL(Ours) | 0.364 | 0.091 | 0.079 | 0.155 | 0.031 | 0.303 | 0.055 | 0.083 | 0.117 | 0.144 | 0.142 |
| Δ | -65% | -83% | -87% | -82% | -88% | -79% | -85% | -84% | -88% | -75% | -80% |

**Experimental settings.** For the prompt learning method, we employ the baseline IVLP (Rasheed et al., 2023) as described in Section 3. Following standard experimental settings (Zhou et al., 2022b; Rasheed et al., 2023), we train a CLIP ViT-B/16 (Radford et al., 2021) on ImageNet in a few-shot fashion, by randomly sampling 16 images per class in training. Under identical training conditions (using the same few-shot training data on ImageNet and hyperparameters), we independently optimize a set of prompt pairs with differences only in their random Gaussian initialization.

**Main results.** As illustrated in Figure 1 (middle), although all trained prompts achieve nearly identical ImageNet (ID) test accuracy, their label prediction distributions of misclassified images on the EuroSAT dataset (OOD) exhibit significant differences. For instance, the first prompt tends to misclassify samples as "Highway or Road" whereas the third prompt tends to categorize them as "Permanent Crop Land." This phenomenon implies that prompts optimized under identical conditions can converge to different local optima, resulting in the divergence in their prediction distributions when significant distribution shifts occur.

**The ubiquity of view bias.** To show that view bias also exists in datasets other than EuroSAT, for every dataset, we compute its Jenson-Shannon (JS) divergence (relative to EuroSAT) between independently-trained prompts' predicted label distributions for misclassified images. As shown in Table 1, the average

JS divergence on most datasets is comparable to EuroSAT, implying the generality of the observation in Figure 1. As another piece of empirical evidence of view bias, we present attention heatmaps of different, independently-trained prompts on the source dataset ImageNet and three distinct target datasets including EuroSAT, Flowers102 (Nilsback & Zisserman, 2008), and StanfordCars (Krause et al., 2013). As shown in Figure 2, the heatmaps of different prompts indeed capture a diverse range of views. Concretely, different prompts exhibit significantly varied attention distributions in both ID and OOD scenarios. For example, on the ImageNet dataset, various prompts focus on different aspects: some on the whole object, some on specific parts of the object, and others on the background areas which may serve as potential cues for classification. Similarly, in the StanfordCars dataset, different prompts highlight different parts of the cars, such as headlights, wheels, and windshields, while some capture more background information.

**Discussion on possible extensions to larger VLMs.** While our main empirical investigations are restricted to CLIP, we expect similar findings may also be obtained in larger models. In particular, a large number of existing VLMs such as LLaVa (Liu et al., 2024), MiniGPT-4 (Zhu et al., 2023) and Instruct-BLIP (Dai et al., 2023) directly leverage CLIP and its derivatives as image encoders. It has been shown that fine-tuning those image encoders can lead to worse OOD performance (Karamcheti et al., 2024), similar to our empirical observation that the OOD accuracy can drop after prompt learning. We thus envision that our results may be extended to those VLMs, but leave rigorous investigations as future work due to computation constraints. Empirically, we also apply our method to SigLIP (Zhai et al., 2023) and observe similar performance gains as to CLIP (see Section 6.7).

## 5 Analysis and Methodology

In this section, we theoretically analyze multi-modal prompt learning with the representation matching objective and characterize the features learned by multi-modal prompts. First, by investigating the role of the softmax-attention mechanism in prompt learning, we show that the representation matching objective can be decomposed into complementary terms that isolate the *feature selection* effect of visual and textual prompts (Section 5.1). Then, we analyze the innate *multi-solution* nature of prompt learning under a linear feature model and further relate this to view bias and OOD generalization failure (Section 5.2). Motivated by our analysis, we then introduce the $M^3PL$ framework, showing that the view bias of single prompt pair can be mitigated by aggregating the output of multiple prompt pairs (Section 5.3), and further propose a cross-modal contrastive regularizer to facilitate the learning of more diverse views in different prompt pairs (Section 5.4).

### 5.1 Prompt Learning as Feature Selection

**Self-attention model.** We begin our analysis by introducing a model of single-head self-attention, which serves as a primary building block of transformers. Concretely, let $\boldsymbol{Z}_{\mathrm{in}} = (\boldsymbol{z}_0, \ldots, \boldsymbol{z}_N)^\top \in \mathbb{R}^{(N+1) \times d_0}$ be the input sequence of the self-attention layer with $\boldsymbol{z}_0$ being the representation token ($\boldsymbol{z}_{\mathrm{cls}}$ on the vision branch and $\boldsymbol{w}_{\mathrm{EOS}}$ on the language branch). The output of the layer is then defined as

$$\boldsymbol{Z}_{\mathrm{out}} = \phi\left(\boldsymbol{Z}_{\mathrm{in}} \boldsymbol{W}_Q \boldsymbol{W}_K^\top \boldsymbol{Z}_{\mathrm{in}}^\top\right) \boldsymbol{Z}_{\mathrm{in}} \boldsymbol{W}, \tag{2}$$

where $\boldsymbol{W}_Q \in \mathbb{R}^{d_0 \times m}$, $\boldsymbol{W}_K \in \mathbb{R}^{d_0 \times m}$ and $\boldsymbol{W} \in \mathbb{R}^{d_0 \times d}$ are model weights, and $\phi$ is a softmax nonlinearity that acts row-wise when taking into a matrix as input. We consider the case where the weights $\boldsymbol{W}_Q \in \mathbb{R}^{d_0 \times m}$, $\boldsymbol{W}_K \in \mathbb{R}^{d_0 \times m}$, and $\boldsymbol{W} \in \mathbb{R}^{d_0 \times d}$ have been pre-trained and keep frozen during prompt learning. The final representation $\boldsymbol{v} \in \mathbb{R}^d$ is then mapped from the representation token in $\boldsymbol{Z}_{\mathrm{out}}$, given by its first row:

$$\boldsymbol{v}^\top = \phi\left(\boldsymbol{z}_0^\top \boldsymbol{W}_Q \boldsymbol{W}_K^\top \boldsymbol{Z}_{\mathrm{in}}^\top\right) \boldsymbol{Z}_{\mathrm{in}} \boldsymbol{W}. \tag{3}$$

**Multi-modal prompt learning.** For simplicity, we consider appending a single learnable prompt token $\boldsymbol{p} \in \mathbb{R}^{d_0}$ to the raw input $\boldsymbol{Z}_{\mathrm{in}}$: let $\widetilde{\boldsymbol{Z}}_{\mathrm{in}} = \begin{bmatrix} \boldsymbol{p}^\top \\ \boldsymbol{Z}_{\mathrm{in}} \end{bmatrix} \in \mathbb{R}^{(N+2) \times d_0}$ be the new input. The new representation for classification is then given by

$$\tilde{\boldsymbol{v}}^\top = \phi\left(\begin{bmatrix} \boldsymbol{z}_0^\top \boldsymbol{W}_Q \boldsymbol{W}_K^\top \boldsymbol{p} & \boldsymbol{z}_0^\top \boldsymbol{W}_Q \boldsymbol{W}_K^\top \boldsymbol{Z}_{\mathrm{in}}^\top \end{bmatrix}\right) \begin{bmatrix} \boldsymbol{p}^\top \boldsymbol{W} \\ \boldsymbol{Z}_{\mathrm{in}} \boldsymbol{W} \end{bmatrix}. \tag{4}$$

Our main observation here is that $\tilde{\boldsymbol{v}}$ is a *weighted mixture* of the raw input $\boldsymbol{Z}_{\text{in}}$ and the prompt $\boldsymbol{p}$. In other words, $\tilde{\boldsymbol{v}}$ takes the form of

$$\tilde{\boldsymbol{v}} = \eta \boldsymbol{v} + (1 - \eta) \boldsymbol{W}^\top \boldsymbol{p}, \tag{5}$$

where the *weighting coefficient* $\eta$ is obtained by expanding and reweighting the original softmax-attention map in Eq. (3), with its concrete form detailed in Appendix B.1. In multi-modal prompt learning, both vision branch and language branch have their learnable prompts. To avoid confusion, in what follows we shall use $\boldsymbol{v}(\tilde{\boldsymbol{v}})$ and $\boldsymbol{t}(\tilde{\boldsymbol{t}})$ to denote visual and textual representations, respectively. For other parameters, we will use subscripts "$v$" and "$t$" to denote if they belong to vision branch or language branch.

**Feature selection effect of prompts.** We then introduce the common representation matching objective in multi-modal prompt learning. For a $C$-way classification problem with training distribution $D$, multi-modal prompt learning aims to minimize

$$\mathcal{L}_{\text{CE}} = \mathbb{E}_{(\boldsymbol{Z}_{\text{in}}, y) \sim D} \left[ -\log \frac{\exp(\text{sim}(\tilde{\boldsymbol{v}}, \tilde{\boldsymbol{t}}_y)/\tau)}{\sum_{i=1}^C \exp(\text{sim}(\tilde{\boldsymbol{v}}, \tilde{\boldsymbol{t}}_i)/\tau)} \right], \tag{6}$$

where for every label $y \in \{1, \ldots, C\}$, $\tilde{\boldsymbol{t}}_y$ denotes the textual representations of $y$, and $\text{sim}(\cdot, \cdot) : \mathbb{R}^d \times \mathbb{R}^d \to \mathbb{R}$ is a similarity measure. In practice, $\text{sim}(\cdot, \cdot)$ is often the cosine similarity as in Eq. (1). In our analysis, we assume $\text{sim}(\cdot, \cdot)$ to be the inner product $\langle \cdot, \cdot \rangle$. Note that inner-product and cosine similarity are equivalent if we normalize the representations before calculating the loss. In practice, normalizing the representations often results in comparable classification performance to using unnormalized representations (Radford et al., 2021). We consider a binary classification setting with $y \in \{-1, 1\}$ and $\tau = 1$. This allows us to derive a cleaner form of the loss function that reveals the role of multi-modal prompts, which is formally shown by Proposition 1.

**Proposition 1** (Objective decomposition). *Under the conditions stated above, we have*

$$\mathcal{L}_{\text{CE}} = \mathbb{E}_{(\boldsymbol{Z}_{\text{in}}, y) \sim D} \log \Big( 1 + \exp \big\{ \underbrace{\eta_v(\eta_{t,y} - \eta_{t,-y})\langle \boldsymbol{W}_t^\top \boldsymbol{p}_t, \boldsymbol{v} \rangle}_{(1)} + \underbrace{(1 - \eta_v)\langle \eta_{t,-y}\boldsymbol{t}_{-y} - \eta_{t,y}\boldsymbol{t}_y, \boldsymbol{W}_v^\top \boldsymbol{p}_v \rangle}_{(2)}$$
$$+ \underbrace{(1 - \eta_v)(\eta_{t,y} - \eta_{t,-y})\langle \boldsymbol{W}_t^\top \boldsymbol{p}_t, \boldsymbol{W}_v^\top \boldsymbol{p}_v \rangle}_{(3)} + \underbrace{\eta_v \langle \eta_{t,-y}\boldsymbol{t}_{-y} - \eta_{t,y}\boldsymbol{t}_y, \boldsymbol{v} \rangle}_{(4)} \big\} \Big), \tag{7}$$

*where $\eta_v$ denotes the weighting coefficient in the vision branch, and $\eta_{t,y}$ denotes the weighting coefficient in the language branch for class $y \in \{-1, 1\}$.*

*Proof.* The complete proofs of Proposition 1 and the following propositions are deferred to Appendix B. $\square$

**Remarks.** Proposition 1 shows that the multi-modal prompt learning objective can be decomposed into terms that reflect the similarity between (1) the textual prompt and the visual representation, (2) the visual prompt and the textual representation, (3) visual and textual prompts, and (4) visual and textual representations. In particular, the first two terms can be viewed as a *feature selection* mechanism that allows the model to emphasize the task-related features in both visual and textual representations by adjusting $\boldsymbol{p}_t$ and $\boldsymbol{p}_v$. This also justifies the advantage of *multi-modal* prompts as it makes the model expressible enough to accommodate distribution shifts in both vision and text domains, which we empirically verify in Secion 6.6.

## 5.2 View Bias and OOD Generalization

**Multi-solution property of prompt learning.** Given Proposition 1, our key insight on multi-modal prompt learning is that minimizing $\mathcal{L}_{\text{CE}}$ can lead to *multiple representation matching schemes* that give *similar training risks*, resulting in the observed view bias of different prompts. As an example, given input $\boldsymbol{Z}_{\text{in}}$ from a class $y \in \{-1, 1\}$, we assume that each input token $\boldsymbol{z}_i$ for $i \in \{0, \ldots, N\}$ is a linear combination of a set of orthogonal, unit-norm features $\boldsymbol{f}_j, j \in \{1, \ldots, l\}$ with each feature $\boldsymbol{f}_j \in \mathbb{R}^d$. Similar assumptions are common in analyzing the feature learning process of neural networks, and prior work has shown that it can capture many practical feature learning characteristics (Allen-Zhu & Li, 2023; Zhang et al., 2024). For

simplicity, we assume that there are no "useless" features, i.e., every feature is correlated with the label on $D$. Then, due to Eq. (3), we can write $\boldsymbol{v}$ as also a combination of those features: $\boldsymbol{v} = \sum_{j=1}^{l} \beta_j \boldsymbol{f}_j$ for some random variable $\beta_j \in \mathbb{R}$ depending on distribution $D$ and pre-trained weights. Hence, by Eq. (7) we have that matching $\boldsymbol{p}_t$ and $\boldsymbol{v}$ by pushing $\boldsymbol{W}_t^\top \boldsymbol{p}_t$ along the direction of *any feature $\boldsymbol{f}_i$ that correlates with the label $y$* can reduce the training risk $\mathcal{L}_{\mathrm{CE}}$. We note that the crux of the above argument exploits the inner-product term between multi-modal representations. Our analysis can thus be extended to other loss functions with inner-product terms such as the sigmoid loss used by SigLIP (Zhai et al., 2023).

Given the multi-solution nature of the objective, how the finally learned prompt $\boldsymbol{p}_t$ correlates with each feature $\boldsymbol{f}_i$ *cannot* be uniquely determined by the training distribution $D$ and the loss function $\mathcal{L}_{\mathrm{CE}}$, but is also determined by the concrete process of optimization, where the inductive biases of the neural network and gradient descent play a critical role. We will detail this next.

**View bias in feature learning.** Recently, several works (Allen-Zhu & Li, 2023; Zhang & Bottou, 2023) show that independent neural networks with the same architecture trained by gradient descent can converge to different optima that each extracts only a *subset* of useful features. This phenomenon also relates to the *simplicity bias* of neural networks (Shah et al., 2020), i.e., neural networks may prefer "simple" solutions, such as using only a subset of useful features for classification, over "complex" solutions, such as using all useful features. In our prompt learning setting, such a bias corresponds to $\langle \boldsymbol{f}_j, \boldsymbol{W}_t^\top \boldsymbol{p}_t \rangle \neq 0$ holding for only $j \in S$ with $S$ being a subset of $\{1, \dots, l\}$, which we name *view bias*. Empirically, we show that such a view bias indeed holds in practice in Section 4. Yet, rigorously proving it is challenging due to the requirement of analyzing fine-grained gradient descent dynamics, which we leave as future work.

While view bias may benefit in-distribution generalization by serving as an implicit regularization, Proposition 2 formally shows that it can adversely *harm* generalization under certain distribution shifts.

**Proposition 2** (View bias can harm generalization)**.** *Under the conditions stated above, consider a test distribution $D'$ satisfying $\mathbb{E}_{D'|y=1}\beta_j = \mathbb{E}_{D'|y=-1}\beta_j$ for every $j \in S$ and $\mathbb{E}_{D'|y=1}\beta_j \neq \mathbb{E}_{D'|y=-1}\beta_j$ for every $j \in S'$ with $S' \subseteq \{1, \dots, l\} \setminus S$. Assume that the weighting coefficients satisfy $\eta_{t,1} = \eta_{t,-1}$. We then have*

$$\mathbf{Pr}_{(\boldsymbol{Z}_{\mathrm{in}}, y) \sim D'}\big[\langle \boldsymbol{v}, \tilde{\boldsymbol{t}}_y \rangle > \langle \boldsymbol{v}, \tilde{\boldsymbol{t}}_{-y} \rangle\big] = \mathbf{Pr}_{(\boldsymbol{Z}_{\mathrm{in}}, y) \sim D'}[\langle \boldsymbol{v}, \boldsymbol{t}_y \rangle > \langle \boldsymbol{v}, \boldsymbol{t}_{-y} \rangle]. \tag{8}$$

**Remarks.** Proposition 2 reflects a distribution shift scenario where only a feature subset $S'$ remains useful in the test distribution $D'$. In an extreme case, if this useful subset $S'$ does not overlap with the feature subset $S$ extracted by prompt learning, then prompt learning would essentially lead to *no improvement* in test accuracy since no additional useful feature is properly conditioned during prompting. To make matters worse, when the learned features $S$ have spurious correlations with labels (Simon, 1954; Schölkopf et al., 2021) or contain large noise, over-reliance on those features by prompt learning may even *decrease* distributional robustness. This is consistent with our empirical observations that in some cases, prompt learning does not improve the performance of CLIP under large distribution shifts and sometimes even decreases it.

### 5.3 M³PL: Multiple Prompt Pairs and View Aggregation

Motivated by the above analysis, this section proposes M³PL that aims to mitigate the intrinsic flaw of view bias in prompt learning by introducing multiple, paired multimodal prompts and aggregating their views.

**Incorporating multiple prompt pairs.** Specifically, building upon the vanilla prompt learning approach in Section 3, for each layer in the first $J$ layers of CLIP's vision branch, we introduce $M$ sets of learnable prompts, denoted as $\boldsymbol{p}_{v,1} = \{\boldsymbol{p}_{v,1}^j\}_{j=0}^{J-1}, \dots, \boldsymbol{p}_{v,M} = \{\boldsymbol{p}_{v,M}^j\}_{j=0}^{J-1}$. Symmetrically, in the language branch, we also add $M$ sets of learnable prompts in the first $J$ layers denoted by $\boldsymbol{p}_{t,i}$ for $i \in \{1, \dots, M\}$. For each $i$, we treat the visual prompt set $\boldsymbol{p}_{v,i}$ and the textual prompt set $\boldsymbol{p}_{t,i}$ as a *prompt pair*. The input sequences for the $i$-th prompt pair and the $j$-th layer in the vision and language branches are then given by

$$\widetilde{\boldsymbol{Z}}_i^j = \{\boldsymbol{p}_{v,i}^j, \boldsymbol{Z}^j\}, \quad \widetilde{\boldsymbol{W}}_i^j = \{\boldsymbol{w}_{\mathrm{SOS}}^j, \boldsymbol{p}_{t,i}^j, \boldsymbol{w}_y^j, \boldsymbol{w}_{\mathrm{EOS}}^j\}. \tag{9}$$

**Paired representation matching with multiple prompts.** In the forward process, we obtain visual and textual representations for all prompt pairs, given by $\tilde{\boldsymbol{v}}_i = f(\widetilde{\boldsymbol{Z}}_i^0; \boldsymbol{\theta}_f, \boldsymbol{p}_{v,i})$ and $\tilde{\boldsymbol{t}}_i = g(\widetilde{\boldsymbol{W}}_i^0; \boldsymbol{\theta}_g, \boldsymbol{p}_{t,i})$ for

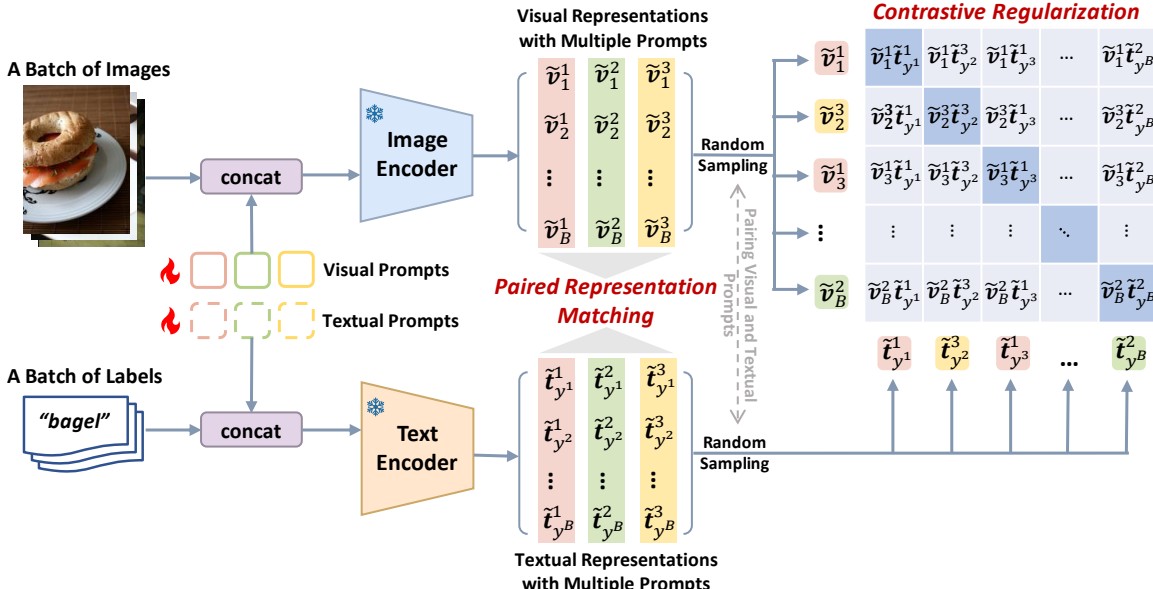

Figure 3: **The M³PL framework.** We introduce multiple paired visual and textual prompts and jointly optimize each prompt pair using representation matching (Section 5.3). Meanwhile, we randomly sample from the multiple prompt representations corresponding to each example for contrastive regularization to further enhance the learning of more diverse prompts (Section 5.4). We use superscripts to denote the indices of prompt pairs in the figure for visual clarity.

the $i$-th prompt pair. During training, we sum the representation matching loss for every prompt pair:

$$\mathcal{L}_{\text{match}} = \sum_{i=1}^{M} \mathbb{E}_{(\boldsymbol{X}, y) \sim D} \left[ -\log \frac{\exp(\cos(\tilde{\boldsymbol{v}}_i, \tilde{\boldsymbol{t}}_{y,i})/\tau)}{\sum_{y'=1}^{C} \exp(\cos(\tilde{\boldsymbol{v}}_i, \tilde{\boldsymbol{t}}_{y',i})/\tau)} \right], \tag{10}$$

where for each $y \in \{1, \ldots, C\}$, $\tilde{\boldsymbol{t}}_{y,i}$ denotes the textual representation corresponding to the label $y$ for the $i$-th prompt pair. During inference, we average the prediction logits obtained from all prompt pairs.

**Exploiting view bias by aggregating different views.** The key intuition of our approach is that as we empirically observe in Figures 1 and 2, *independently trained prompts tend to have distinct view biases.* Hence, aggregating them naturally results in a richer collection of useful features. Formally, Proposition 3 demonstrates that if independently optimized prompts extract independent feature subsets, then aggregating them by simply averaging their representation matching scores can benefit OOD generalization.

**Proposition 3** (Effectiveness of aggregating multiple views)**.** *Under the same conditions as in Proposition 2, consider $M$ prompts that each independently extracts a feature subset $S_i, i \in \{1, \ldots, M\}$ with $|S_i| = s$ and the elements in each $S_i$ uniformly drawn from $\{1, \ldots, l\}$. We then have*

$$\mathbf{Pr}_{(\boldsymbol{Z}_{\text{in}}, y) \sim D'} \left[ \sum_{i=1}^{M} \langle \boldsymbol{v}, \tilde{\boldsymbol{t}}_{y,i} \rangle > \sum_{i=1}^{M} \langle \boldsymbol{v}, \tilde{\boldsymbol{t}}_{-y,i} \rangle \right] > \mathbf{Pr}_{(\boldsymbol{Z}_{\text{in}}, y) \sim D'} [\langle \boldsymbol{v}, \boldsymbol{t}_y \rangle > \langle \boldsymbol{v}, \boldsymbol{t}_{-y} \rangle] \tag{11}$$

*with probability at least $1 - \Theta\left( \left( \frac{l - |S'|}{l} \right)^{sM} \right)$.*

**Remarks.** Proposition 3 assumes a scenario where different prompts learn independent views, while only some of them remain useful in OOD data. Since we cannot determine which views are useful solely based on ID data, simply aggregating all of them seems to be a fair approach as adopted by M³PL. Yet, such aggregation may also induce redundant views, which is indeed observed in our experiments (see Appendix C.4.1).

Thus, it would be more efficient to *actively* incentivize different prompts to learn more diverse views. In the next section, we propose cross-modal contrastive regularization to achieve this goal.

### 5.4 Cross-Modal Contrastive Regularization

To further enhance the diversity of the learned views of different prompts, we introduce a cross-modal contrastive regularization penalty. The main idea is to *maximize* the representation difference between *different* prompt pairs while matching the representations in the same prompt pair. Concretely, given a batch of $B$ examples $\{(\boldsymbol{X}^1, y^1), \ldots, (\boldsymbol{X}^B, y^B)\}$, for every example, we randomly sample a prompt pair $\{\boldsymbol{p}_{v,r(i)}, \boldsymbol{p}_{t,r(i)}\}$, where $r(i) \in \{1, \ldots, M\}$ denotes the prompt pair index for the $i$-th example in the batch. We then calculate the cross-model contrastive regularization penalty by

$$\mathcal{L}_{\text{contrast}} = \sum_{i=1}^{B} \left[ -\log \frac{\exp(\cos(\tilde{\boldsymbol{v}}_{i,r(i)}, \tilde{\boldsymbol{t}}_{y^i,r(i)})/\tau)}{\sum_{k=1}^{B} \exp(\cos(\tilde{\boldsymbol{v}}_{i,r(i)}, \tilde{\boldsymbol{t}}_{y^k,r(k)}/\tau)} - \log \frac{\exp(\cos(\tilde{\boldsymbol{v}}_{i,r(i)}, \tilde{\boldsymbol{t}}_{y^i,r(i)}/\tau)}{\sum_{k=1}^{B} \exp(\cos(\tilde{\boldsymbol{v}}_{k,r(k)}, \tilde{\boldsymbol{t}}_{y^i,r(i)}/\tau)} \right], \qquad (12)$$

Our overall training objective is then given by:

$$\mathcal{L} = \mathcal{L}_{\text{match}} + \lambda \mathcal{L}_{\text{contrast}}, \qquad (13)$$

where $\lambda > 0$ is the balancing coefficient.

**Explanation on contrastive regularization.** Contrastive loss aims to pull positive examples together and push negative examples apart. Here, we treat the visual and textual representations of the *same input* with the *same prompt pair* as positive examples and *all* other cases as negative examples. In other words, representations with different prompts would become *negative* examples and are thus pushed apart, even for the same input. This relates to the effect of "class collision" that have been observed in the contrastive learning literature (Goyal et al., 2023). However, instead of mitigating this effect, we actively *leverage* it to encourage different prompt pairs to learn more diverse views. We empirically verify this in Secion 6.6.

## 6 Experiments

### 6.1 Protocols for Evaluating Generalization Performance

In prompt learning, the average accuracy on OOD test sets is commonly used to evaluate a model's generalization performance (Zhou et al., 2022a; Khattak et al., 2023a;b). However, Taori et al. (2020) points out that OOD accuracy is insufficient to reflect the accuracy drop under distribution shifts after fine-tuning. And Miller et al. (2021) finds through large-scale experiments that there is a strong correlation between a model's OOD and ID performance, suggesting that improvements in OOD accuracy cannot be entirely attributed to the fine-tuning methods. Instead, it may simply be due to better fit on the ID distribution. Therefore, to comprehensively evaluate the generalization performance of prompt learning methods, we propose the *effective robustness ratio*, inspired by Taori et al. (2020), as a complementary metric to average OOD accuracy. Its expression is as follows:

$$\rho(f) = \frac{\overline{acc}_{ood}(f) - \overline{acc}_{ood}(f_0)}{\overline{acc}_{id}(f) - \overline{acc}_{id}(f_0)} \qquad (14)$$

where $f_0$ is the zero-shot CLIP, $f$ is the fine-tuned model, and $\overline{acc}_{ood}(\cdot)$ denotes the average OOD accuracy.

**Discussion on effective robustness ratio.** This metric measures the relative accuracy drop under distribution shifts for the fine-tuned model compared to the pre-trained CLIP. Generally, $\rho(f) \leq 0\%$ indicates that the model has overfitted to the ID distribution. For $\rho(f) \in (0\%, 100\%)$, the larger $\rho(f)$, the smaller the compromise of fine-tuning methods on the generalization ability of CLIP and the greater the generalization ability. In particular, when the ID and OOD distributions are nearly identical, $\rho(f)$ approaches $100\%$.

## 6.2 Experimental Settings

**Base-to-new generalization.** This setting validates the model's capacity to generalize unseen classes during fine-tuning. We equally divided the classes of each dataset into new and base sections. After training on base classes, the model is directly zero-shot tested on new classes.

**Cross-dataset generalization.** To verify the generalization performance of our method when both the vision and language modalities distributions shift during testing, we fine-tune the model on ImageNet and then conduct zero-shot testing directly on other downstream datasets.

**Domain generalization.** Unlike the previous settings, which exhibit significant distribution shifts in both vision and language modalities, DG shows distribution shifts only in the vision modality and is not the main focus of our method. Nonetheless, our proposed M$^3$PL still achieves comparable performance in the DG setting, slightly surpassing previous prompt-based algorithms. Details are provided in Appendix C.3.3.

**Implementation details.** Following MaPLe (Khattak et al., 2023a), we employ the ViT-B/16 based CLIP as the backbone. We use a few-shot setting that samples 16 shots per class and report the results averaged over three runs. For M$^3$PL, we use a normal distribution with a mean of zero to randomly initialize the prompts, and increase the variance with the number of prompts ($M$) to ensure diversity. In our experiments, we set $M$ to 8. Due to the use of $\mathcal{L}_{\text{contrast}}$, we use a larger batch size while reducing the training iterations to compensate for the computation overhead. Since ProDA lacks an official implementation, we report the results in Derakhshani et al. (2023) for the Base-to-New setting. For other baselines, we reproduce the results based on the provided hyperparameters. Please refer to Appendix C.1 for additional training details.

**Datasets.** For cross-dataset generalization and from base-to-new generalization settings, we follow the protocols of Zhou et al. (2022a;b); Khattak et al. (2023a) and consider 11 recognition datasets, including ImageNet (Deng et al., 2009) and Caltech101 (Fei-Fei et al., 2004) for generic recognition, OxfordPets (Parkhi et al., 2012), StanfordCars (Krause et al., 2013), Flowers102 (Nilsback & Zisserman, 2008), Food101 (Bossard et al., 2014) and FGVCAircraft (Maji et al., 2013) for fine-grained classification, SUN397 (Xiao et al., 2010) for scene classification, DTD (Cimpoi et al., 2014) for texture recognition, EuroSAT (Helber et al., 2019) for satellite image recognition, and UCF101 (Soomro et al., 2012) for action recognition .

**Baselines.** We use zero-shot CLIP (Radford et al., 2021), CoOp (Zhou et al., 2022b), CoCoOp (Zhou et al., 2022a), ProDA (Lu et al., 2022), MaPLe (Khattak et al., 2023a), and PromptSRC (Khattak et al., 2023b).

## 6.3 Base-to-New Generalization

In the generalization from base to new classes, shifts in both modalities occur due to partial observations during fine-tuning. In Table 2, M$^3$PL demonstrates superior performance across all average metrics on 11 datasets, comprising base and new class test accuracy, harmonic mean accuracy, and effective robustness ratio. In tests on new classes, M$^3$PL consistently outperforms the state-of-the-art PromptSRC in 9/11 datasets, improving the average accuracy by 1.05% without compromising base class accuracy. It is worth mentioning that on the larger-scale dataset ImageNet, M$^3$PL surpasses PromptSRC by **1.13%** in zero-shot new class test accuracy. Full results are detailed in Appendix C.3.1.

## 6.4 Cross-Dataset Generalization

Table 3 illustrates that M$^3$PL substantially improves both the average zero-shot test accuracy and the effective robustness ratio in the cross-dataset generalization setting with shifts in both vision and language modalities. Compared to zero-shot CLIP, existing methods only achieve a modest increase of 0.61% in average accuracy, whereas M$^3$PL realizes a substantial improvement of 2.16%. Even excluding the superior performance on the EuroSAT dataset, where accuracy increased by 9.12% compared to zero-shot CLIP, M$^3$PL still demonstrates an average accuracy gain of 1.39%. Against the state-of-the-art PromptSRC, M$^3$PL excels in 8/10 target datasets, boosting the effective robustness ratio by *2.8 times without markedly affecting ID performance*. These results highlight the exceptional robustness of our framework in handling distribution shifts. Full results are detailed in Appendix C.3.2.

Table 2: Comparison with previous methods in base-to-new generalization. All baselines are reproduced with reported parameters. HM and $\rho(f)$ refer to harmonic mean and effective robustness ratio, respectively.

| Dataset | | Zero-shot CLIP | CoOp | CoCoOp | ProDA | MaPLe | PromptSRC | M³PL(Ours) |
|---|---|---|---|---|---|---|---|---|
| **Average on 11 datasets** | Base | 69.48 | 82.29 | 80.49 | 81.56 | 82.00 | 84.18 | **84.90** |
| | New | 74.28 | 68.78 | 72.04 | 72.30 | 74.88 | 75.76 | **76.81** |
| | HM | 71.80 | 74.93 | 76.03 | 76.65 | 78.28 | 79.75 | **80.65** |
| | $\rho(f)$ | - | -43% | -20% | -16% | 5% | 10% | **16%** |
| ImageNet | Base | 72.37 | 76.47 | 75.93 | 75.40 | 76.87 | **77.80** | 77.72 |
| | New | 68.10 | 67.50 | 70.13 | 70.23 | 70.73 | 70.60 | **71.73** |
| | HM | 70.17 | 71.71 | 72.91 | 72.72 | 73.67 | 74.03 | **74.60** |
| Caltech101 | Base | 97.22 | 98.10 | 97.80 | 98.27 | 97.93 | 98.10 | **98.45** |
| | New | 94.21 | 93.20 | 93.00 | 93.23 | **95.23** | 94.03 | 94.25 |
| | HM | 95.69 | 95.59 | 95.34 | 95.68 | **96.56** | 96.02 | 96.30 |
| OxfordPets | Base | 91.23 | 94.53 | 95.03 | 95.43 | 95.60 | 95.33 | **95.85** |
| | New | 97.20 | 95.80 | 97.73 | 97.83 | 98.00 | 97.27 | **98.15** |
| | HM | 94.12 | 95.16 | 96.36 | 96.62 | 96.79 | 96.29 | **96.99** |
| StanfordCars | Base | 63.69 | 75.60 | 70.73 | 74.70 | 72.40 | 78.13 | **79.07** |
| | New | 74.92 | 70.03 | 72.50 | 71.20 | 73.67 | **75.37** | 74.03 |
| | HM | 68.85 | 72.71 | 71.60 | 72.91 | 73.03 | **76.73** | 76.47 |
| Flowers102 | Base | 71.70 | 97.53 | 94.43 | 97.70 | 96.10 | **98.17** | **98.17** |
| | New | **77.52** | 71.30 | 70.63 | 68.68 | 72.87 | 77.37 | 75.72 |
| | HM | 74.50 | 82.38 | 80.81 | 80.66 | 82.89 | **86.54** | 85.50 |
| Food101 | Base | 90.07 | 89.50 | 90.57 | 90.30 | 90.83 | 90.63 | **90.85** |
| | New | 91.17 | 88.90 | 91.27 | 88.57 | 92.03 | 91.47 | **92.08** |
| | HM | 90.62 | 89.20 | 90.92 | 89.43 | 91.43 | 91.05 | **91.46** |
| FGVCAircraft | Base | 27.55 | 38.67 | 35.33 | 36.90 | 36.17 | 42.27 | **42.88** |
| | New | 35.93 | 29.80 | 31.07 | 34.13 | 34.87 | 37.43 | **39.11** |
| | HM | 31.19 | 33.66 | 33.06 | 35.46 | 35.51 | 39.70 | **40.91** |
| SUN397 | Base | 69.38 | 81.20 | 79.37 | 78.67 | 80.97 | 82.77 | **82.84** |
| | New | 75.58 | 70.43 | 76.23 | 76.93 | 78.30 | 78.50 | **78.85** |
| | HM | 72.35 | 75.43 | 77.77 | 77.79 | 79.61 | 80.58 | **80.80** |
| DTD | Base | 53.13 | 79.67 | 76.93 | 80.67 | 80.47 | 82.97 | **83.91** |
| | New | 60.27 | 49.37 | 54.67 | 56.48 | 58.40 | 59.57 | **61.75** |
| | HM | 56.48 | 60.96 | 63.92 | 66.44 | 67.68 | 69.35 | **71.14** |
| EuroSAT | Base | 56.98 | 88.97 | 87.10 | 83.90 | 91.07 | 92.70 | **96.72** |
| | New | 63.74 | 56.00 | 61.87 | 66.00 | 72.83 | 73.17 | **77.94** |
| | HM | 60.17 | 68.74 | 72.35 | 73.88 | 80.94 | 81.79 | **86.32** |
| UCF101 | Base | 70.99 | 85.00 | 82.13 | 85.23 | 83.57 | 87.10 | **87.40** |
| | New | 78.47 | 64.20 | 73.30 | 71.97 | 76.73 | 78.57 | **81.29** |
| | HM | 74.54 | 73.15 | 77.46 | 78.04 | 80.00 | 82.62 | **84.23** |

Table 3: Comparison with previous methods in the cross-dataset generalization. All baselines are reproduced with reported parameters. **M³PL shows a significant improvement in the effective robustness ratio**.

| | Source | Target | | | | | | | | | | | |
|---|---|---|---|---|---|---|---|---|---|---|---|---|---|
| | ImageNet | Caltech 101 | Oxford Pets | Stanford Cars | Flowers 102 | Food101 | FGVC Aircraft | SUN397 | DTD | EuroSAT | UCF101 | **Average** | $\rho(f)$ |
| Zero-shot CLIP | 66.68 | 93.31 | 89.10 | 65.51 | 70.73 | 85.88 | **24.66** | 62.60 | 44.09 | 48.40 | 67.59 | 65.19 | - |
| CoOp | **71.63** | 93.73 | 88.27 | 63.63 | 68.70 | 85.63 | 19.27 | 64.60 | 41.63 | 48.33 | 67.37 | 64.12 | -22% |
| CoCoOp | 71.20 | **94.43** | 90.60 | 64.83 | 71.03 | 86.13 | 23.23 | 67.20 | **45.87** | 41.00 | 68.50 | 65.28 | 2% |
| MaPLe | 70.40 | 93.53 | 90.03 | 64.90 | 71.93 | 85.97 | 23.77 | 66.67 | 45.03 | 44.87 | 67.47 | 65.42 | 6% |
| PromptSRC | 71.37 | 93.37 | 90.30 | 65.70 | 70.43 | 86.47 | 23.57 | 67.43 | 45.83 | 45.43 | **69.50** | 65.80 | 13% |
| M³PL(Ours) | 71.11 | 93.94 | **90.88** | **66.35** | **72.26** | **86.56** | 24.23 | **67.65** | 45.61 | **57.52** | 68.45 | **67.35** | **49%** |

Table 4: **Ablation** on each component of M$^3$PL framework in the cross-dataset generalization setting.

| Method | ID Acc. | Average OOD Acc. | $\rho(f)$ |
|---|---|---|---|
| Zero-shot CLIP | 66.68 | 65.19 | - |
| Independent V-L Prompting (IVLP) | 71.16 | 65.22 | 1% |
| + textual multi-prompts | 71.00 | 66.15 | 22% |
| + visual multi-prompts (w/o matching) | 70.99 | 66.64 | 34% |
| + visual multi-prompts | 70.95 | 66.87 | 39% |
| + $\mathcal{L}_{\text{contrast}}$ (w/o collision) | 70.96 | 66.82 | 38% |
| + $\mathcal{L}_{\text{contrast}}$ | **71.11** | **67.35** | **49%** |

## 6.5 Performance Analysis

While the empirical evidence in Section 4 implies the prevalence of view bias in prompt learning, our proposed M$^3$PL algorithm, which leverages view bias, yields varying degrees of improvement across different datasets in the cross-dataset generalization setting. This section provides an in-depth analysis of this phenomenon.

**Theoretical interpretation.** As shown by Proposition 1, minimizing the representation matching objective can be viewed as implementing a *feature selection* mechanism for both visual and textual pre-trained features. Hence, the degree of improvement of M$^3$PL on a specific dataset depends on not only prompt learning but also the overall quality and adaptability of CLIP's pre-trained features and features that are learnable in downstream ID data. For example, if pre-trained features are not predictive or the downstream ID data lacks predictive features under distribution shifts, prompt learning may not improve the OOD performance much.

Empirically, to examine the quality and adaptability of pre-trained and ID features, we design two complementary metrics. (1) **Informativeness:** the generalization potential of CLIP's pre-trained features on a specific target dataset, measured by the average performance of zero-shot CLIP and the linear probe on CLIP's features on this dataset. (2) **Transferability:** the distributional similarity between target datasets and ImageNet, measured by the average of the cosine similarity between visual and textual representations of examples from the two datasets. We then examine the linear relationship between those metrics and M$^3$PL's performance gains compared to zero-shot CLIP. More details are provided in Appendix D.

**Results.** On the EuroSAT dataset, where CLIP's pre-trained features' capability and the visual-textual joint distribution similarity are both high, only M$^3$PL fully realizes the above theoretical potential. Conversely, on the FGVCAircraft dataset, both metrics are lower, resulting in poor prompt learning performance. Nevertheless, M$^3$PL *still performs best* among existing prompt-based methods. Performance on other datasets can also be explained by these two metrics. For a detailed analysis, refer to Appendix D.3.

**Reduction in JS Divergence.** Moreover, we report in Table 1 the relative average JS divergence between the predicted label distributions of aggregated prompts trained with different random seeds and initializations using M$^3$PL. Comparing it to the results without M$^3$PL, we observe that the average JS divergence significantly decreases in all datasets, with an average reduction rate of **80%**. This further demonstrates the effectiveness of M$^3$PL in mitigating the view bias problem.

## 6.6 Ablation Study

**Effectiveness of M$^3$PL.** As shown in Table 4, the baseline IVLP, impaired by view bias, shows negligible improvement in term of OOD accuracy over zero-shot CLIP. In contrast, integrating multiple prompts significantly enhances OOD accuracy (rows 2-4), supporting our theoretical analysis in Section 5.3. The incorporation of visual prompts further improves model performance (rows 3-4), corroborating our analysis of the multi-modal prompt learning objective in Section 5.1. The introduction of the matching design in $\mathcal{L}_{\text{multi}}$ (row 4) proves more effective than scenarios where interplay exists among different prompt pairs (row 3). In addition, without $\mathcal{L}_{\text{contrast}}$, the OOD accuracy of M$^3$PL is already 1.68% higher than CLIP, indicating that it effectively exploits view bias and enhances generalization under distribution shifts. Please refer to Appendix C.4.1 for an ablation study on the number of prompts.

**Cross-modal contrastive regularization.** In Table 4, we further delineate the contributions of our proposed cross-modal contrastive regularization. Compared to scenarios with class collisions (row 6), avoiding collisions (row 5) does not enhance generalization performance with the addition of the $\mathcal{L}_{\text{contrast}}$. This observation aligns with our discussion in Section 5.4, indicating that our proposed regularization objective effectively steers different prompts to learn diverse views, thereby improving OOD performance. We also conducted additional ablation studies on the impact of the $\lambda$ and batch size in Appendix C.4.2.

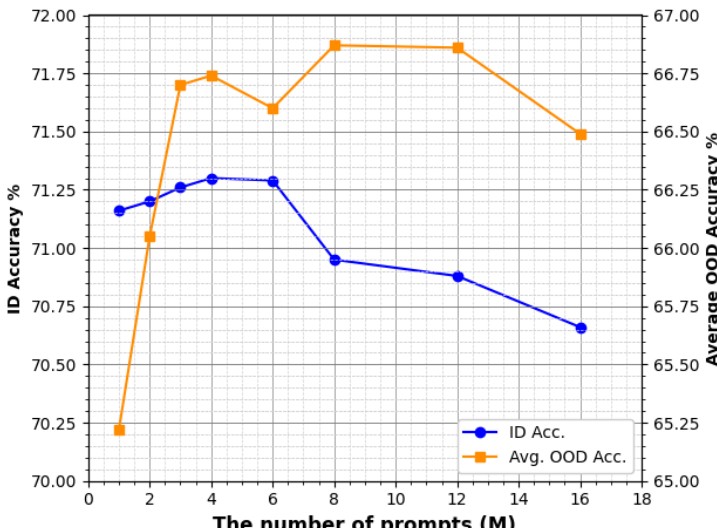

Figure 4: **Ablation on the number of prompts in the cross-dataset generalization setting.** The left vertical axis represents the ID test accuracy on ImageNet, and the right vertical axis indicates the average zero-shot OOD test accuracy across target datasets. The trends of ID accuracy and average OOD accuracy with the number of prompts $M$ ($\lambda = 0$) are depicted by curves with circular and square markers, respectively.

**The number of prompts.** Based on the ID and OOD performance with different numbers of prompts on the held-out validation set, we set the number of prompts ($M$) to 8 in the experiments to achieve optimal OOD accuracy while maintaining satisfactory ID accuracy. Figure 5 presents the ID and OOD test accuracy curves as $M$ varies in the cross-dataset generalization setting. With an increase in $M$, ID accuracy initially increases slightly but then decreases due to feature redundancy. In contrast, OOD accuracy significantly improves first and then gradually declines. This behavior is attributed to the lack of view diversity when $M$ is small, and feature redundancy becomes dominant and causes performance degradation when $M$ is large.

**Computational overhead.** MaPLe sets the prompt length for both visual and language branches to half that of uni-modal prompt learning methods like CoOp, resulting in total floating-point operations (FLOPs) comparable to CoOp (Khattak et al., 2023a). Following the same prompt length and depth settings as MaPLe, the FLOPs of M³PL are approximately $M$x those of CoOp. Since M³PL requires learning multiple prompt pairs, it incurs a higher time cost. To address this, we reduce the number of training iterations, ensuring that the total training time for M³PL is comparable to that of PromptSRC. Table 5 presents a practical comparison between our M³PL and previous prompt learning methods in terms of training time and inference speed. The training time for all methods is measured under the base-to-new generalization setting, using a single Nvidia A100 GPU on the SUN397 dataset for complete training. Compared to the state-of-the-art PromptSRC, our M³PL model reduces training time by approximately 5% when $M = 8$, but at the expense of a 44% decrease in inference speed. Notably, as shown in Figure 5, with $M = 4$, M³PL already achieves significant improvements over existing methods. Therefore, using fewer prompts can be an option, which sacrifices part of generalization performance for a large increase in computational efficiency.

**Comparison to temporal prompt aggregation in PromptSRC.** Please refer to Appendix C.4.3.

**Prompt length and depth.** Please refer to Appendix C.4.4.

|  | CoOp | CoCoOp | MaPLe | PromptSRC | M$^3$PL (Ours, $M = 8$) |
|---|---|---|---|---|---|
| Training time (min) | 8.52 | 56.23 | 6.17 | 22.85 | 21.77 |
| Inference time (images/s) | 323.1 | 19.0 | 328.2 | 329.7 | 183.4 |

Table 5: The practical training and inference time comparison with previous methods ($M = 8$).

### 6.7 More Results on SigLIP

We extend our experiments to the new state-of-the-art vision-language pre-training model, SigLIP (Zhai et al., 2023), which uses pairwise sigmoid loss. The results demonstrate that M$^3$PL similarly enhances the generalization performance of fine-tuned SigLIP, validating the scalability and universality of our approach as a robust prompt learning method for large multi-modal models. For detailed experimental results, please refer to Appendix C.5.

## 7 Discussion

**Limitations.** M$^3$PL adopts a straightforward aggregation strategy of averaging different prompts' logit scores. While being simple and empirically effective, this design choice may lead to suboptimal generalization on specific OOD tasks due to feature redundancy. Additionally, our experiments are currently limited to CLIP/SigLIP ViT-B/16 and visual recognition tasks, although we expect that our results may also hold for other backbones as well.

**Future work.** The currently rapidly evolving test-time prompt tuning methods (Shu et al., 2022) could potentially serve as an effective means to filter the optimal prompts learned by M$^3$PL. Furthermore, our method has the potential to extend to larger-scale VLMs and more diverse tasks. We hope that M$^3$PL, a theoretically grounded, highly scalable, and minimally constrained framework, will establish itself as a universal baseline of regularized prompt learning methods and facilitate future research in this domain.

### Broader Impact Statement

This work is devoted to developing more robust ways to fine-tune VLMs. Therefore, it may benefit the broad research area of building machine learning models that are robust, generalizable, and trustworthy. However, it may also inherit the negative societal impact of the original VLMs, such as potential misuse cases, biased output, and privacy and security concerns.

### Acknowledgments

This work was supported in part by the National Key Research and Development Program of China under STI 2030-Major Projects 2021ZD0200300, and in part by the National Natural Science Foundation of China under Grant 62176133.

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

## A  Appendix

In the appendix, we provide additional supplementary information, including proofs of the theoretical derivations presented in the main text, further implementation details, comprehensive experimental results, additional ablation studies, and visualization outcomes. The specific organization of the appendix sections is as follows:

- Proofs of Theoretical Results (Appendix B)

    - Proof of Eq. (5) (Appendix B.1)
    - Proof of proposition 1 (Appendix B.2)
    - Proof of proposition 2 (Appendix B.3)
    - Proof of proposition 3 (Appendix B.4)

- Additional Experiment Details (Appendix C)

    - Implementation details (Appendix C.1)
    - Datasets (Appendix C.2)
    - Full results (Appendix C.3)
    - Additional ablation study (Appendix C.4)

- Supplementary Experiments (Appendix D)

    - Linear Probe Experiment (Appendix D.1)
    - Datasets Representation Similarity Experiment (Appendix D.2)
    - Multivariate Linear Regression (Appendix D.3)

## B  Proofs of Theoretical Results

### B.1  Proof of Eq. (5)

*Proof.* Define $\alpha_i = \frac{\exp(\boldsymbol{z}_0^\top \boldsymbol{W}_Q \boldsymbol{W}_K^\top \boldsymbol{z}_i)}{\sum_{j=0}^N \exp(\boldsymbol{z}_0^\top \boldsymbol{W}_Q \boldsymbol{W}_K^\top \boldsymbol{z}_j)} \; \forall i \in \{0, \ldots, N\}$, $\alpha_{\boldsymbol{p}} = \frac{\exp(\boldsymbol{z}_0^\top \boldsymbol{W}_Q \boldsymbol{W}_K^\top \boldsymbol{p})}{\sum_{j=0}^N \exp(\boldsymbol{z}_0^\top \boldsymbol{W}_Q \boldsymbol{W}_K^\top \boldsymbol{z}_j)}$, and the weighting

coefficient $\eta = \frac{\sum_{j=0}^N \exp(\boldsymbol{z}_0^\top \boldsymbol{W}_Q \boldsymbol{W}_K^\top \boldsymbol{z}_j)}{\exp(\boldsymbol{z}_0^\top \boldsymbol{W}_Q \boldsymbol{W}_K^\top \boldsymbol{p}) + \sum_{j=0}^N \exp(\boldsymbol{z}_0^\top \boldsymbol{W}_Q \boldsymbol{W}_K^\top \boldsymbol{z}_j)}$. Applying the definition of $\alpha_i$ to Eq. (3) gives

$$\boldsymbol{v} = \boldsymbol{W}^\top \sum_{i=0}^N \alpha_i \boldsymbol{z}_i. \tag{15}$$

Plugging the above equations into Eq. (4) then gives

$$\tilde{\boldsymbol{v}} = \boldsymbol{W}^\top \sum_{i=0}^N \eta \alpha_i \boldsymbol{z}_i + \boldsymbol{W}^\top \eta \alpha_{\boldsymbol{p}} \boldsymbol{p} \tag{16}$$
$$= \eta \boldsymbol{v} + (1 - \eta) \boldsymbol{W}^\top \boldsymbol{p}$$

as desired. $\qquad\square$

### B.2  Proof of Proposition 1

*Proof.* Applying Eq. (5) to both vision and language branches gives

$$\tilde{\boldsymbol{v}} = \eta_v \boldsymbol{v} + (1 - \eta_v) \boldsymbol{W}_v^\top \boldsymbol{p}_v \tag{17}$$

and

$$\tilde{\boldsymbol{t}}_y = \eta_{t,y} \boldsymbol{t}_y + (1 - \eta_{t,y}) \boldsymbol{W}_t^\top \boldsymbol{p}_t, \; \forall y \in \{-1, 1\}. \tag{18}$$

Recall that the cross-entropy loss with $\text{sim}(\cdot, \cdot)$ being the inner product, $C = 2$ and $\tau = 1$ can be written as

$$
\begin{aligned}
\mathcal{L}_{\text{CE}} &= \mathbb{E}_{(\boldsymbol{Z}_{\text{in}}, y) \sim D} \left[ - \log \frac{\exp(\langle \tilde{\boldsymbol{v}}, \tilde{\boldsymbol{t}}_y \rangle))}{\exp(\langle \tilde{\boldsymbol{v}}, \tilde{\boldsymbol{t}}_y \rangle) + \exp(\langle \tilde{\boldsymbol{v}}, \tilde{\boldsymbol{t}}_{-y} \rangle)} \right] \\
&= \mathbb{E}_{(\boldsymbol{Z}_{\text{in}}, y) \sim D} \left[ - \log \left( 1 + \exp(\langle \tilde{\boldsymbol{v}}, \tilde{\boldsymbol{t}}_{-y} \rangle - \langle \tilde{\boldsymbol{v}}, \tilde{\boldsymbol{t}}_y \rangle) \right) \right],
\end{aligned}
\tag{19}
$$

where

$$
\begin{aligned}
&\langle \tilde{\boldsymbol{v}}, \tilde{\boldsymbol{t}}_{-y} \rangle - \langle \tilde{\boldsymbol{v}}, \tilde{\boldsymbol{t}}_y \rangle \\
&= \langle \eta_v \boldsymbol{v} + (1 - \eta_v) \boldsymbol{W}_v^\top \boldsymbol{p}_v, \eta_{t,-y} \boldsymbol{t}_{-y} - \eta_{t,y} \boldsymbol{t}_y \rangle + \langle \eta_v \boldsymbol{v} + (1 - \eta_v) \boldsymbol{W}_v^\top \boldsymbol{p}_v, (\eta_{t,y} - \eta_{t,-y}) \boldsymbol{W}_t^\top \boldsymbol{p}_t \rangle \\
&= \eta_v \langle \eta_{t,-y} \boldsymbol{t}_{-y} - \eta_{t,y} \boldsymbol{t}_y, \boldsymbol{v} \rangle + \eta_v (\eta_{t,y} - \eta_{t,-y}) \langle \boldsymbol{v}, \boldsymbol{W}_t^\top \boldsymbol{p}_t \rangle \\
&\quad + (1 - \eta_v) \langle \eta_{t,-y} \boldsymbol{t}_{-y} - \eta_{t,y} \boldsymbol{t}_y, \boldsymbol{W}_v^\top \boldsymbol{p}_v \rangle + (1 - \eta_v)(\eta_{t,y} - \eta_{t,-y}) \langle \boldsymbol{W}_v^\top \boldsymbol{p}_v, \boldsymbol{W}_t^\top \boldsymbol{p}_t \rangle.
\end{aligned}
\tag{20}
$$

Finally, plugging Eq. (20) into Eq. (19) completes the proof. $\qquad\square$

### B.3 Proof of Proposition 2

*Proof.* Applying Eq. (5) to the language branch gives

$$
\tilde{\boldsymbol{t}}_y = \eta_{t,y} \boldsymbol{t}_y + (1 - \eta_{t,y}) \boldsymbol{W}_t^\top \boldsymbol{p}_t, \ \forall y \in \{-1, 1\}.
\tag{21}
$$

Combining the above equation and

$$
\boldsymbol{v} = \sum_{j=1}^l \beta_j \boldsymbol{f}_j, \ \forall y \in \{-1, 1\}
\tag{22}
$$

yields that for every $y$,

$$
\begin{aligned}
\langle \boldsymbol{v}, \tilde{\boldsymbol{t}}_y \rangle &= \sum_{j=1}^l \beta_j \langle \boldsymbol{f}_j, \tilde{\boldsymbol{t}}_y \rangle \\
&= \sum_{j=1}^l \beta_j \left( \eta_{t,y} \langle \boldsymbol{f}_j, \boldsymbol{t}_y \rangle + (1 - \eta_{t,y}) \langle \boldsymbol{f}_j, \boldsymbol{W}_t^\top \boldsymbol{p}_t \rangle \right).
\end{aligned}
\tag{23}
$$

Since $\langle \boldsymbol{f}_j, \boldsymbol{W}_t^\top \boldsymbol{p}_t \rangle = 0$ for every $j \notin S$, we have for every $y$ that

$$
\langle \boldsymbol{v}, \tilde{\boldsymbol{t}}_y \rangle = \eta_{t,y} \sum_{j=1}^l \beta_j \langle \boldsymbol{f}_j, \boldsymbol{t}_y \rangle + (1 - \eta_{t,y}) \sum_{j \in S} \beta_j \langle \boldsymbol{f}_j, \boldsymbol{W}_t^\top \boldsymbol{p}_t \rangle.
\tag{24}
$$

Since for every $j \in S$, the random variable $\beta_j$ satisfies $\mathbb{E}_{D'|y=1} \beta_j = \mathbb{E}_{D'|y=-1} \beta_j$, we have for every $y$ that

$$
\begin{aligned}
\mathbb{E}_{(\boldsymbol{Z}_{\text{in}}, y) \sim D'} \langle \boldsymbol{v}, \tilde{\boldsymbol{t}}_y \rangle &= \eta_{t,y} \mathbb{E}_{(\boldsymbol{Z}_{\text{in}}, y) \sim D'} \sum_{j=1}^l \beta_j \langle \boldsymbol{f}_j, \boldsymbol{t}_y \rangle + C_1 \\
&= \eta_{t,y} \mathbb{E}_{(\boldsymbol{Z}_{\text{in}}, y) \sim D'} \langle \boldsymbol{v}, \boldsymbol{t}_y \rangle + C_1,
\end{aligned}
\tag{25}
$$

where $C_1$ is a constant that does not depend on $y$. Using the assumption that $\eta_{t,1} = \eta_{t,2}$) we then have

$$
\begin{aligned}
&\mathbf{Pr}_{(\boldsymbol{Z}_{\text{in}}, y) \sim D'} \left[ \langle \boldsymbol{v}, \tilde{\boldsymbol{t}}_y \rangle > \langle \boldsymbol{v}, \tilde{\boldsymbol{t}}_{-y} \rangle \right] \\
&= \mathbf{Pr}_{(\boldsymbol{Z}_{\text{in}}, y) \sim D'} [\eta_{t,y} \langle \boldsymbol{v}, \boldsymbol{t}_y \rangle + C_1 > \eta_{t,-y} \langle \boldsymbol{v}, \boldsymbol{t}_{-y} \rangle + C_1] \\
&= \mathbf{Pr}_{(\boldsymbol{Z}_{\text{in}}, y) \sim D'} [\langle \boldsymbol{v}, \boldsymbol{t}_y \rangle > \langle \boldsymbol{v}, \boldsymbol{t}_{-y} \rangle].
\end{aligned}
\tag{26}
$$

This completes the proof. $\qquad\square$

**Remark 1.** In Proposition 2, we make the assumption that the coefficients $\eta_{t,1}$ and $\eta_{t,2}$ are equal in order to simplify our analysis. However, in practice, this assumption may not hold since those coefficients also depend on the pre-trained attention weights as well as the learned prompts. We posit that this discrepancy may lead to the scenarios where single-prompt learning can also harm OOD generalization due to the overfitting of $\eta_{t,y}$ to the training distribution, as we empirically observed in the experiments.

**Remark 2.** In our analysis, the learnable prompts in the vision and language branches have a symmetric structure. Therefore, the analysis in Proposition 2 can be directly extended to the setting where both the vision and language branches have learnable prompts.

### B.4 Proof of Proposition 3

*Proof.* Akin to the proof of Proposition 2, we have for every $y$ that

$$\langle \boldsymbol{v}, \tilde{\boldsymbol{t}}_y \rangle = \eta_{t,y} \sum_{j=1}^{l} \beta_j \langle \boldsymbol{f}_j, \boldsymbol{t}_y \rangle + (1 - \eta_{t,y}) \sum_{j \in S} \beta_j \langle \boldsymbol{f}_j, \boldsymbol{W}_t^\top \boldsymbol{p}_t \rangle. \tag{27}$$

Proposition 2 indicates that for every $i$, if the $i$-th prompt extracts a feature subset $S_i, i \in \{1, \ldots, M\}$ with $S_i \cap S' = \varnothing$, then

$$\begin{aligned}
&\mathbf{Pr}_{(\boldsymbol{z}_{\text{in}}, y) \sim D'}\left[\langle \boldsymbol{v}, \tilde{\boldsymbol{t}}_{y,i} \rangle > \langle \boldsymbol{v}, \tilde{\boldsymbol{t}}_{-y,i} \rangle\right] \\
&= \mathbf{Pr}_{(\boldsymbol{z}_{\text{in}}, y) \sim D'}\left[\langle \boldsymbol{v}, \boldsymbol{t}_{y,i} \rangle > \langle \boldsymbol{v}, \boldsymbol{t}_{-y,i} \rangle\right]
\end{aligned} \tag{28}$$

for this prompt. Conversely, if $S_i \cap S' \neq \varnothing$, then the second term in the RHS of Eq. (27) also depends on the label $y$, which leads to better accuracy since it brings more expressibility by incorporating more predictive features (note that $\beta_j$ also depends on the prompts). We thus have

$$\begin{aligned}
&\mathbf{Pr}_{(\boldsymbol{z}_{\text{in}}, y) \sim D'}\left[\langle \boldsymbol{v}, \tilde{\boldsymbol{t}}_{y,i} \rangle > \langle \boldsymbol{v}, \tilde{\boldsymbol{t}}_{-y,i} \rangle\right] \\
&> \mathbf{Pr}_{(\boldsymbol{z}_{\text{in}}, y) \sim D'}\left[\langle \boldsymbol{v}, \boldsymbol{t}_{y,i} \rangle > \langle \boldsymbol{v}, \boldsymbol{t}_{-y,i} \rangle\right]
\end{aligned} \tag{29}$$

for this prompt. Therefore, as long as at least one prompt satisfies $S_i \cap S' \neq \varnothing$, we must have

$$\begin{aligned}
&\mathbf{Pr}_{(\boldsymbol{z}_{\text{in}}, y) \sim D'}\left[\sum_{i=1}^{M} \langle \boldsymbol{v}, \tilde{\boldsymbol{t}}_{y,i} \rangle > \sum_{i=1}^{M} \langle \boldsymbol{v}, \tilde{\boldsymbol{t}}_{-y,i} \rangle\right] \\
&> \mathbf{Pr}_{(\boldsymbol{z}_{\text{in}}, y) \sim D'}\left[\langle \boldsymbol{v}, \boldsymbol{t}_y \rangle > \langle \boldsymbol{v}, \boldsymbol{t}_{-y} \rangle\right].
\end{aligned} \tag{30}$$

We then formally characterize the above probability (Denoting the event that the inequality 30 holds by $E$). Since here we work with the simple case that different prompts extract independent feature subsets that are uniformly drawn from $\{1, \ldots, l\}$, the probability that at least one prompt extracts features in $S'$ is given by the union bound over $M$ Bernoulli distributions, each with failure probability $p = \frac{C_{l-|S'|}^s}{C_l^s}$:

$$\begin{aligned}
\mathbf{Pr}(E) &= 1 - p^M \\
&= 1 - \left(\frac{C_{l-|S'|}^s}{C_l^s}\right)^M.
\end{aligned} \tag{31}$$

Consider the case where $l$ is sufficiently large, we have $C_l^s = \Theta(l^s)$ and $C_{l-|S'|}^s = \Theta((l - |S'|)^s)$. Plugging them into Eq. (31) gives the desired result. $\qquad \square$

## C Additional Experiment Details

### C.1 Implementation Details

**Base-to-new generalization.** Following the settings of MaPLe, we set the prompt depth $J$ to 9, and the length of both visual and textual prompts to 2. Due to the significant impact of class collision on the

effectiveness of our proposed cross-modal contrastive regularization objective, and its frequency being related to the ratio of batch size to the number of classes in a dataset, we employ varying batch sizes across different datasets to maintain this ratio around 0.6 (e.g., 32 for DTD (Cimpoi et al., 2014), 64 for UCF101 (Soomro et al., 2012)). We utilize an SGD optimizer with a learning rate of 2.5e-3, weight decay of 5e-4, and training for 30 epochs (for a few datasets prone to overfitting, the training was limited to 20 epochs). The number of prompts, $M$, is set to 8, with a balance coefficient, $\lambda$, of 1.0 (and 1.2 for EuroSAT). For computing the effective robustness ratio, we use the average zero-shot test accuracy of new classes across 11 datasets as $\overline{acc}_{ood}(\cdot)$, and the average test accuracy of base classes as $\overline{acc}_{id}(\cdot)$.

**Cross-dataset generalization.** Following the settings of MaPLe (Khattak et al., 2023a), we set the prompt depth $J$ to 3, with both visual and textual prompts having a length of 2. Due to the incorporation of a cross-modal contrastive regularization objective, we utilize a larger batch size of 512. To compensate for the additional time expenditure, all the models are trained for only 50 epochs (1550 iterations, compared to MaPLe's 20,000 iterations). The model is optimized using SGD with a learning rate of 2.5e-3 and a weight decay of 5e-4. We set the number of prompts $M$ to 8 and the balancing coefficient $\lambda$ to 1.0. For calculating the effective robustness ratio ($\rho(f)$), we use the average zero-shot test accuracy on the target dataset as $\overline{acc}_{ood}(\cdot)$, and the test accuracy on ImageNet as $\overline{acc}_{id}(\cdot)$.

**Domain generalization.** Same as the cross-dataset generalization, we set the prompt depth, $J$, to be 3, with both visual and textual prompts having a length of 2. We employ the SGD optimizer with a learning rate of $2.5 \times 10^{-3}$, weight decay of $5 \times 10^{-4}$, and a batch size of 512, training the model for 50 epochs. The number of prompts, $M$, is set to 8, and the balancing parameter, $\lambda$, is 0.1. For the effective robustness ratio, we use the average zero-shot test accuracy on the target dataset as $\overline{acc}_{ood}(\cdot)$ and the test accuracy on ImageNet as $\overline{acc}_{id}(\cdot)$.

**Reproducibility.** We provide publicly the source code of M$^3$PL, which contains the configuration files we used, to ensure the reliability and reproducibility of our experimental results. All experiments are conducted on NVIDIA A100 GPUs.

## C.2 Datasets

**ImageNet** (Deng et al., 2009): The ImageNet dataset contains over 14 million high-resolution images, manually annotated, and categorized into 1000 classes. It is widely used for image classification and object detection tasks.

**Caltech101** (Fei-Fei et al., 2004): The Caltech101 dataset includes 101 object categories and 1 background category, with 9k images. The number of images per category ranges from 40 to 800, with an average of about 50 images per category.

**OxfordPets** (Parkhi et al., 2012): The OxfordPets dataset comprises 7349 images of cats and dogs, divided into 37 categories. Each category contains approximately 200 images, suitable for pet recognition and classification tasks.

**StanfordCars** (Krause et al., 2013): The StanfordCars dataset consists of 16,185 images of cars, categorized into 196 classes. Each class represents a specific car model and manufacturing year, primarily used for car classification and recognition tasks.

**Flowers102** (Nilsback & Zisserman, 2008): The Flowers102 dataset includes 8189 images of flowers, categorized into 102 classes. The number of images per category ranges from 40 to 258, with an average of about 80.

**Food101** (Bossard et al., 2014): The Food101 dataset comprises 101,000 images of 101 food categories. Each category contains 1000 images, used for food recognition and classification tasks.

**FGVCAircraft** (Maji et al., 2013): The FGVCAircraft dataset contains 10,000 images of aircraft, categorized into 100 classes with 100 images per category, most of which are airplanes.

**SUN397** (Xiao et al., 2010): The SUN397 dataset includes 108,753 images of scenes, categorized into 397 classes. The number of images per category ranges from 100 to 2000, with an average of about 300.

**DTD** (Cimpoi et al., 2014): The Describable Textures Dataset (DTD) comprises 5640 images of textures, divided into 47 categories. Each category contains 120 images, used for texture recognition and classification tasks.

**EuroSAT** (Helber et al., 2019): The EuroSAT dataset contains 27,000 satellite images, categorized into 10 classes. Each class contains 2000 to 3000 images, primarily used for geospatial classification tasks.

**UCF101** (Soomro et al., 2012): The UCF101 dataset includes 13,320 video clips categorized into 101 action classes. Each class represents a specific sport or action, mainly used for action recognition tasks.

### C.3 Full Results

In this section, we report the average accuracy and standard deviation from three runs with three different random seeds in three generalization benchmarks. It is important to note that all baselines are **reproduced** using the official configuration file parameters on the same random seeds and hardware as our experiments, ensuring fairness in comparison.

#### C.3.1 Base-to-New Generalization

The full experimental results in the base-to-new generalization setting are shown in Table 6. Please note that, due to the absence of an official implementation for ProDA (Lu et al., 2022), we report only the results provided by Derakhshani et al. (2023) in Table 2 of the main text, and do not include the full results in the appendix.

#### C.3.2 Cross-Dataset Generalization

The full experimental results in the cross-dataset generalization setting are shown in Table 7.

#### C.3.3 Domain Generalization

**Datasets.** For DG, we use four ImageNet-derived datasets with different domain shifts: ImageNetV2 (Recht et al., 2019), ImageNet-Sketch (Wang et al., 2019), ImageNet-A (Hendrycks et al., 2021b), and ImageNet-R (Hendrycks et al., 2021a).

The full experimental results in the domain generalization setting are shown in Table 8. Unlike the previous two settings, in the DG setting, only the visual modality experiences shifts. Although existing methods have achieved commendable results in this scenario, our $M^3PL$ still attains enhancements in both the average target dataset accuracy and the effective robustness ratio.

### C.4 Additional Ablation Study

#### C.4.1 Effectiveness of $M^3PL$

Figure 5 presents the variation curves of both in-distribution (ID) and out-of-distribution (OOD) accuracy with the changing number of prompts ($M$) in the cross-dataset setting. The ID accuracy initially increases slightly with an increase in $M$ and then decreases, aligning with our analysis in Section 1. This trend is attributed to the view bias bias of prompts. When $M$ is relatively small, the aggregation of useful features from different views enhances ID test accuracy. However, as $M$ further increases, redundant features exacerbate overfitting. In contrast, the OOD accuracy significantly rises before gradually decreasing. This is because when $M$ is too small, the insufficient variety of views leads to inadequate coverage of OOD predictive features, leading to a rapid improvement in OOD performance as $M$ increases. But with a larger $M$, the dominance of redundant and irrelevant features from the introduced views deteriorates the performance. Considering the trends in both ID and OOD changes, we opt for $M = 8$ to trade off performance with computational cost.

| Dataset | | Zero-shot CLIP | CoOp | CoCoOp | MaPLe | PromptSRC | M³PL(Ours) | Δ |
|---|---|---|---|---|---|---|---|---|
| **Average on 11 datasets** | Base | 69.48 | 82.29 | 80.49 | 82.00 | 84.18 | **84.90** | +0.72 |
| | New | 74.28 | 68.78 | 72.04 | 74.88 | 75.76 | **76.81** | +1.05 |
| | HM | 71.80 | 74.93 | 76.03 | 78.28 | 79.75 | **80.65** | +0.90 |
| | $\rho(f)$ | - | -43% | -20% | 5% | 10% | **16%** | +6% |
| ImageNet | Base | 72.37±0.00 | 76.47±0.17 | 75.93±0.24 | 76.87±0.05 | **77.80**±0.00 | 77.72±0.08 | -0.08 |
| | New | 68.10±0.00 | 67.50±0.22 | 70.13±0.33 | 70.73±0.33 | 70.60±0.08 | **71.73**±0.04 | +1.13 |
| | HM | 70.17 | 71.71 | 72.91 | 73.67 | 74.03 | **74.60** | +0.57 |
| Caltech101 | Base | 97.22±0.00 | 98.10±0.00 | 97.80±0.08 | 97.93±0.12 | 98.10±0.16 | **98.45**±0.05 | +0.35 |
| | New | 94.21±0.00 | 93.20±0.41 | 93.00±0.29 | **95.23**±0.21 | 94.03±0.19 | 94.25±0.22 | +0.22 |
| | HM | 95.69 | 95.59 | 95.34 | **96.56** | 96.02 | 96.30 | +0.28 |
| OxfordPets | Base | 91.23±0.00 | 94.53±0.38 | 95.03±0.40 | 95.60±0.22 | 95.33±0.09 | **95.85**±0.35 | +0.52 |
| | New | 97.20±0.00 | 95.80±0.99 | 97.73±0.09 | 98.00±0.36 | 97.27±0.48 | **98.15**±0.16 | +0.83 |
| | HM | 94.12 | 95.16 | 96.36 | 96.79 | 96.29 | **96.99** | +0.70 |
| StanfordCars | Base | 63.69±0.00 | 75.60±1.13 | 70.73±0.71 | 72.40±0.29 | 78.13±0.25 | **79.07**±0.62 | +0.94 |
| | New | 74.92±0.00 | 70.03±0.62 | 72.50±0.86 | 73.67±0.60 | **75.37**±0.33 | 74.03±0.52 | -1.34 |
| | HM | 68.85 | 72.71 | 71.60 | 73.03 | **76.73** | 76.47 | -0.26 |
| Flowers102 | Base | 71.70±0.00 | 97.53±0.09 | 94.43±0.66 | 96.10±0.22 | **98.17**±0.05 | **98.17**±0.20 | +0.00 |
| | New | **77.52**±0.00 | 71.30±1.87 | 70.63±0.98 | 72.87±0.98 | 77.37±0.17 | 75.72±0.93 | -1.65 |
| | HM | 74.50 | 82.38 | 80.81 | 82.89 | **86.54** | 85.50 | -1.04 |
| Food101 | Base | 90.07±0.00 | 89.50±0.45 | 90.57±0.09 | 90.83±0.05 | 90.63±0.12 | **90.85**±0.10 | +0.22 |
| | New | 91.17±0.00 | 88.90±0.45 | 91.27±0.47 | 92.03±0.12 | 91.47±0.12 | **92.08**±0.28 | +0.61 |
| | HM | 90.62 | 89.20 | 90.92 | 91.43 | 91.05 | **91.46** | +0.41 |
| FGVCAircraft | Base | 27.55±0.00 | 38.67±0.12 | 35.33±0.97 | 36.17±0.09 | 42.27±0.54 | **42.88**±0.84 | +0.61 |
| | New | 35.93±0.00 | 29.80±0.43 | 31.07±0.21 | 34.87±1.68 | 37.43±0.73 | **39.11**±0.32 | +2.56 |
| | HM | 31.19 | 33.66 | 33.06 | 35.51 | 39.70 | **40.91** | +1.21 |
| SUN397 | Base | 69.38±0.00 | 81.20±0.08 | 79.37±0.48 | 80.97±0.25 | 82.77±0.09 | **82.84**±0.22 | +0.07 |
| | New | 75.58±0.00 | 70.43±1.65 | 76.23±0.58 | 78.30±0.41 | 78.50±0.57 | **78.85**±0.24 | +0.35 |
| | HM | 72.35 | 75.43 | 77.77 | 79.61 | 80.58 | **80.80** | +0.22 |
| DTD | Base | 53.13±0.00 | 79.67±0.54 | 76.93±0.86 | 80.47±1.38 | 82.97±0.90 | **83.91**±0.29 | +0.94 |
| | New | 60.27±0.00 | 49.37±3.55 | 54.67±4.49 | 58.40±0.64 | 59.57±3.27 | **61.75**±1.70 | +2.18 |
| | HM | 56.48 | 60.96 | 63.92 | 67.68 | 69.35 | **71.14** | +1.79 |
| EuroSAT | Base | 56.98±0.00 | 88.97±1.07 | 87.10±0.70 | 91.07±3.76 | 92.70±0.99 | **96.72**±0.39 | +4.02 |
| | New | 63.74±0.00 | 56.00±3.35 | 61.87±11.47 | 72.83±3.20 | 73.17±3.20 | **77.94**±1.57 | +4.77 |
| | HM | 60.17 | 68.74 | 72.35 | 80.94 | 81.79 | **86.32** | +4.53 |
| UCF101 | Base | 70.99±0.00 | 85.00±0.28 | 82.13±0.17 | 83.57±0.68 | 87.10±0.22 | **87.40**±0.30 | +0.30 |
| | New | 78.47±0.00 | 64.20±3.71 | 73.30±0.85 | 76.73±1.53 | 78.57±1.55 | **81.29**±0.79 | +2.72 |
| | HM | 74.54 | 73.15 | 77.46 | 80.00 | 82.62 | **84.23** | +1.61 |

Table 6: **Full results in the base-to-new generalization setting.** All baseline results are **reproduced** using the reported parameters. The harmonic mean of the base and new class test accuracy is denoted as HM. Improvements over PromptSRC are in blue.

| | Source | Target | | | | | | | | | | | |
|---|---|---|---|---|---|---|---|---|---|---|---|---|---|
| | ImageNet | Caltech101 | OxfordPets | StanfordCars | Flowers102 | Food101 | FGVCAircraft | SUN397 | DTD | EuroSAT | UCF101 | **Average** | $\rho(f)$ |
| Zero-shot CLIP | 66.68±0.00 | 93.31±0.00 | 89.10±0.00 | 65.51±0.00 | 70.73±0.00 | 85.88±0.00 | **24.66**±0.00 | 62.60±0.00 | 44.09±0.00 | 48.40±0.00 | 67.59±0.00 | 65.19 | - |
| CoOp | **71.63**±0.17 | 93.73±0.19 | 88.27±0.52 | 63.63±1.36 | 68.70±0.92 | 85.63±0.12 | 19.27±1.19 | 64.60±0.29 | 41.63±0.62 | 48.33±0.40 | 67.37±0.77 | 64.12 | -22% |
| CoCoOp | 71.20±0.00 | **94.43**±0.05 | 90.60±0.08 | 64.83±0.68 | 71.03±0.87 | 86.13±0.05 | 23.23±0.12 | 67.20±0.08 | **45.87**±0.62 | 41.00±3.22 | 68.50±0.57 | 65.28 | 2% |
| MaPLe | 70.40±0.14 | 93.53±0.54 | 90.03±0.31 | 64.90±0.51 | 71.93±1.27 | 85.97±0.21 | 23.77±0.52 | 66.67±0.21 | 45.03±1.32 | 44.87±2.70 | 67.47±0.37 | 65.42 | 6% |
| PromptSRC | 71.37±0.09 | 93.37±0.19 | 90.30±0.08 | 65.70±0.36 | 70.43±0.24 | 86.47±0.05 | 23.57±0.78 | 67.43±0.25 | 45.83±0.12 | 45.43±1.47 | **69.50**±0.64 | 65.80 | 13% |
| M³PL(Ours) | 71.11±0.02 | 93.94±0.14 | **90.88**±0.09 | **66.35**±0.44 | **72.26**±0.18 | **86.56**±0.06 | 24.23±0.49 | **67.65**±0.06 | 45.61±0.17 | **57.52**±1.82 | 68.45±0.33 | **67.35** | 49% |

Table 7: **Full results in the cross-dataset generalization setting.** All baseline results are **reproduced** using the reported parameters. Since CLIP is directly zero-shot tested without training, its standard deviation is reported as zero.

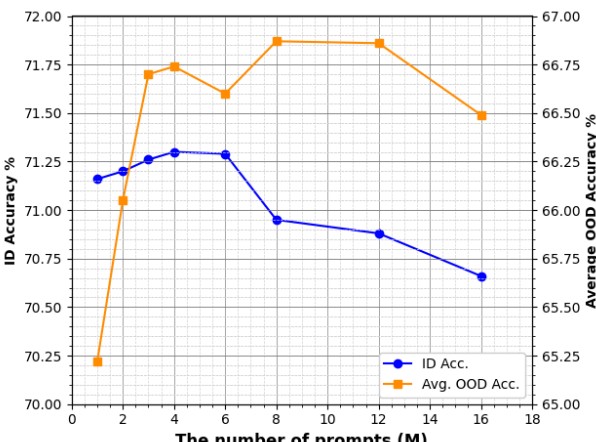

Figure 5: **The ablation experiments on the number of prompts in the cross-dataset generalization setting.** The left vertical axis represents the ID test accuracy on ImageNet, and the right vertical axis indicates the average zero-shot OOD test accuracy across target datasets. The trends of ID accuracy and average OOD accuracy with the number of prompts $M$ ($\lambda = 0$) are depicted by curves with circular and square markers, respectively.

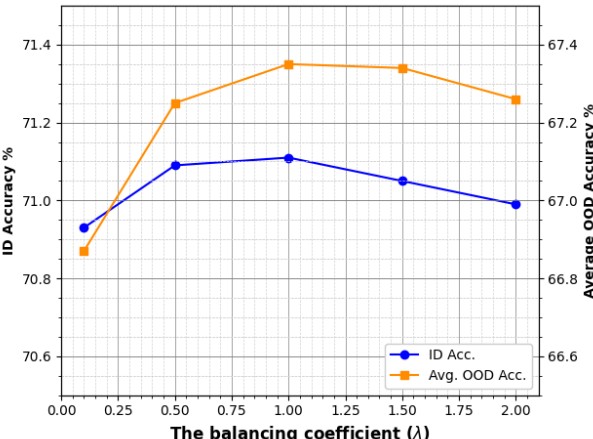

Figure 6: **The ablation experiments on the balancing coefficient $\lambda$ in the cross-dataset generalization setting.** The left vertical axis represents the ID test accuracy on ImageNet, and the right vertical axis indicates the average zero-shot OOD test accuracy across target datasets. The trends of ID accuracy and average OOD accuracy with the balancing coefficient $\lambda$ ($M = 8$) are depicted by curves with circular and square markers, respectively.

| | Source | Target | | | | | |
|---|---|---|---|---|---|---|---|
| | ImageNet | ImageNetV2 | ImageNet-Sketch | ImageNet-A | ImageNet-R | **Average** | $\rho(f)$ |
| Zero-shot CLIP | 66.68±0.00 | 60.91±0.00 | 46.09±0.00 | 47.76±0.00 | 73.97±0.00 | 57.18 | - |
| CoOp | **71.63**±0.17 | 64.27±0.17 | 47.93±0.29 | 50.37±0.25 | 75.33±0.21 | 59.48 | 46% |
| CoCoOp | 71.20±0.00 | 64.27±0.25 | 48.67±0.25 | 50.73±0.19 | 76.10±0.16 | 59.94 | 61% |
| MaPLe | 70.40±0.14 | 63.73±0.12 | 48.60±0.16 | 50.20±0.37 | 76.57±0.12 | 59.78 | 70% |
| PromptSRC | 71.37±0.09 | 64.43±0.05 | 49.53±0.05 | 50.77±0.26 | **77.77**±0.05 | 60.63 | 73% |
| M$^3$PL(Ours) | 70.95±0.07 | **64.49**±0.07 | **49.60**±0.10 | **51.47**±0.09 | 77.40±0.07 | **60.74** | **83%** |

Table 8: **Full results in the domain generalization setting ($\lambda = 0.1$).** All baseline results are **reproduced** using the reported parameters. Since CLIP is directly zero-shot tested without training, its standard deviation is reported as zero.

### C.4.2 Cross-modal Contrastive Regularization

Figure 6 presents the variation curves of ID and OOD accuracy in the cross-dataset generalization setting as a function of the balancing coefficient $\lambda$. Both ID and OOD accuracies initially increase and then decrease with the rising values of $\lambda$. We set $\lambda = 1.0$ to optimally balance $\mathcal{L}_{\text{multi}}$ and $\mathcal{L}_{\text{contrast}}$.

| Batch size | $\lambda$ | ID Acc. | Average OOD Acc. | $\rho(f)$ |
|---|---|---|---|---|
| 512 | 0.0 | 70.95 | 66.87 | 0.39 |
| 128 | 1.0 | 71.02 | 67.15 | 0.45 |
| 256 | 1.0 | 71.11 | 67.25 | 0.47 |
| 512 | 1.0 | 71.11 | 67.35 | 0.49 |

Table 9: **The ablation experiments on the impact of batch size on the cross-modal contrastive regularization objective** in the cross-dataset generalization settings ($M = 8$). Note that under different batch size settings, we control the same iteration for training.

Table 9 presents the ablation study results on the impact of batch size on the cross-modal contrastive regularization objective in the cross-dataset generalization setting. We observe that with $\lambda = 1.0$, the average OOD test accuracy on the target datasets increases as the batch size increases. This trend is attributed to a lower class collision rate when the batch size is smaller, which in turn reduces the performance of the cross-modal contrastive regularization. However, it is still evident that even with a batch size of 128 (feasible on a single Nvidia A100 GPU), there is a significant improvement over the baseline that does not utilize $\mathcal{L}_{\text{contrast}}$, further demonstrating the effectiveness of our proposed regularization objective.

### C.4.3 Comparison to Temporal Prompt Aggregation in PromptSRC

| Method | ID Acc. | Average OOD Acc. | $\rho(f)$ |
|---|---|---|---|
| Independent V-L Prompting (IVLP) | 71.16 | 65.22 | 0.01 |
| IVLP + GPA of PromptSRC | **72.10** | 65.51 (+0.29) | 0.06 |
| IVLP + M$^3$PL (w/o $\mathcal{L}_{\text{contrast}}$) | 70.95 | **66.87** (+1.65) | **0.39** |

Table 10: **Ablation** on aggregation strategy in the cross-dataset setting.

Rather than use *multiple* prompt pairs and aggregate their results, PromptSRC (Khattak et al., 2023b) employs Gaussian weighted prompt aggregation (GPA), which temporally aggregates the results of a *single* prompt pair across its training trajectory. Here we compare the effectiveness of the two techniques in Table 10. As shown in the table, GPA yields little improvement due to the same view obtained from a single optimization trajectory, which is consistent with our analysis that a single optimization trajectory may fail to capture a broad range of views.

### C.4.4    Prompt Length and Depth

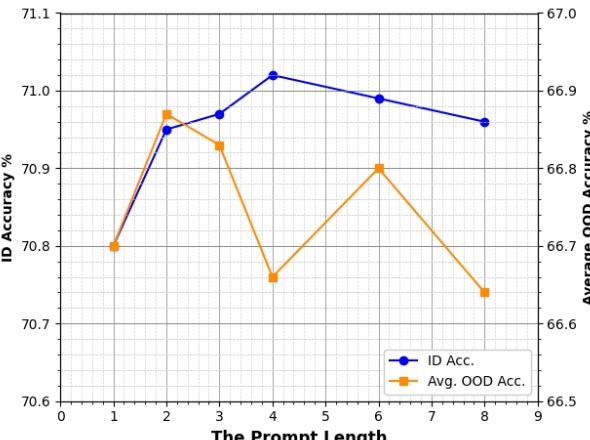

Figure 7: **The ablation experiments on the prompt length in the cross-dataset generalization setting** ($M = 8$, $\lambda = 0$, $J = 3$). The left vertical axis represents the ID test accuracy on ImageNet, and the right vertical axis indicates the average zero-shot OOD test accuracy across target datasets. The trends of ID accuracy and average OOD accuracy with the prompt length are depicted by curves with circular and square markers, respectively.

Figure 7 displays the results of ablation experiments on prompt length in a cross-dataset generalization setting. The results indicate that both ID and OOD test accuracies generally exhibit an initial increase followed by a decrease. Consequently, we select a prompt length of 2 to trade off the performance between ID and OOD scenarios.

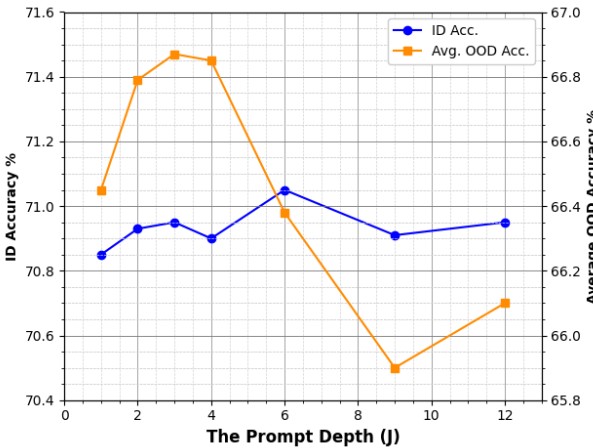

Figure 8: **The ablation experiments on the prompt depth in the cross-dataset generalization setting** ($M = 8$, $\lambda = 0$, and prompt length 2). The left vertical axis represents the ID test accuracy on ImageNet, and the right vertical axis indicates the average zero-shot OOD test accuracy across target datasets. The trends of ID accuracy and average OOD accuracy with the prompt depth $J$ are depicted by curves with circular and square markers, respectively.

Figure 8 presents the ablation study results regarding the depth of prompts $J$ in the cross-dataset generalization. It is observed that both ID and OOD test accuracies generally follow an initial increase followed by a decrease, with OOD test accuracy being more significantly influenced by $J$. We opted for $J = 3$ to trade off the performance between ID and OOD settings.

### C.5 Full Results on SigLIP

Our experimental design in the main text primarily focuses on CLIP. In this section, we report additional results on the latest state-of-the-art image-text pre-trained model, SigLIP (Zhai et al., 2023), to further validate our method's scalability and generalization ability. Specifically, we use the pre-trained SigLIP-B/16 as the backbone and use the same hyperparameters as in our CLIP experiments. We evaluate the performance of the zero-shot SigLIP model, Independent V-L Prompting (IVLP), and M³PL under the standard Base-to-New generalization setting, as shown in Table 11. The results demonstrate that compared to the baseline method IVLP, M³PL achieves a significant improvement in out-of-distribution (OOD) performance (+8.03%) while maintaining in-distribution (ID) performance, corroborating the theoretical analysis in Section 5.2.

Table 11: Results of extending the M³PL approach to the SigLIP (Zhai et al., 2023) backbone in base-to-new generalization. HM and $\rho(f)$ refer to harmonic mean and effective robustness ratio, respectively.

| Dataset | | Zero-shot SigLIP | SigLIP + IVLP | SigLIP + M³PL(Ours) |
|---|---|---|---|---|
| **Average on 11 datasets** | Base | 77.75 | **87.00** | 86.84 |
| | New | 81.23 | 78.75 | **82.61** |
| | HM | 79.45 | 82.67 | **84.67** |
| | $\rho(f)$ | - | -27% | **15%** |
| ImageNet | Base | 79.60 | 81.26 | **81.70** |
| | New | 76.90 | 76.09 | **78.32** |
| | HM | 78.23 | 78.59 | **79.97** |
| Caltech101 | Base | 98.50 | **98.90** | 98.71 |
| | New | 97.80 | 97.38 | **98.36** |
| | HM | 98.15 | 98.13 | **98.53** |
| OxfordPets | Base | 96.50 | 95.27 | **96.97** |
| | New | 98.10 | 98.55 | **98.60** |
| | HM | 97.29 | 96.88 | **97.78** |
| StanfordCars | Base | 85.00 | **91.58** | 89.63 |
| | New | 96.20 | 95.67 | **96.56** |
| | HM | 90.25 | **93.58** | 92.97 |
| Flowers102 | Base | 91.00 | **99.43** | 98.67 |
| | New | 86.00 | 81.06 | **86.24** |
| | HM | 88.43 | 89.31 | **92.04** |
| Food101 | Base | 92.30 | 91.57 | **93.24** |
| | New | 93.60 | 92.38 | **94.15** |
| | HM | 92.95 | 91.97 | **93.69** |
| FGVCAircraft | Base | 29.20 | **48.44** | 41.24 |
| | New | 40.80 | 37.19 | **46.07** |
| | HM | 34.04 | 42.08 | **43.52** |
| SUN397 | Base | 75.30 | 82.69 | **83.75** |
| | New | 78.60 | 76.68 | **81.23** |
| | HM | 76.91 | 79.57 | **82.47** |
| DTD | Base | 75.60 | 88.19 | **88.43** |
| | New | **73.60** | 67.27 | 73.55 |
| | HM | 74.59 | 76.32 | **80.31** |
| EuroSAT | Base | 56.40 | 93.21 | **96.50** |
| | New | 68.00 | 68.64 | **71.51** |
| | HM | 61.66 | 79.06 | **82.15** |
| UCF101 | Base | 75.90 | **86.45** | 86.35 |
| | New | 83.90 | 75.39 | **84.10** |
| | HM | 79.70 | 80.54 | **85.21** |

## D Supplementary Experiments

To further elucidate the performance gain of M³PL in zero-shot cross-dataset generalization across datasets with varying distributions, we conduct two supplementary experiments: (1) A Few-shot Linear Probe experiment based on the same protocol in Radford et al. (2021); Zhou et al. (2022b) to evaluate the generalization

potential of CLIP's pre-trained features on each target dataset, as detailed in Appendix D.1. (2) An assessment of the cosine similarity between average visual and textual representations across target dataset categories and those from ImageNet categories to measure the information gain through few-shot prompt learning on ImageNet, presented in Appendix D.2. For simplicity, we limit our experiments to datasets with zero-shot CLIP's accuracy below 85%. We assume that if zero-shot CLIP performs above this threshold, its pre-trained features are already generally sufficient for generalizing to the target dataset, so the influence of view bias is relatively minor.

Based on the metrics from the above experiments, we use a simple multivariate linear regression model to interpret M$^3$PL's relative improvements over zero-shot CLIP across different datasets, with specific results detailed in Appendix D.3.

### D.1 Linear Probe Experiment

| Method | StanfordCars | Flowers102 | FGVCAircraft | SUN397 | DTD | EuroSAT | UCF101 |
|---|---|---|---|---|---|---|---|
| Zero-shot CLIP | 65.51 | 70.73 | 24.66 | 62.60 | 44.09 | 48.40 | 67.59 |
| Linear Probe CLIP | 80.60 | 97.28 | 83.30 | 73.15 | 70.15 | 86.33 | 82.66 |
| Average Performance | 73.06 | 84.01 | 35.49 | 67.88 | 57.12 | 67.37 | 75.13 |

Table 12: **Few-shot linear probe performance (%) on the target datasets**.

**Experimental settings.** We adhere to the few-shot linear probe setup in Zhou et al. (2022b), sampling 16 instances per class and reporting the average results across three random seeds. Consistent with the cross-dataset generalization setting discussed in Section 6, we employ ViT-B/16 as the backbone for CLIP.

Results in Table 12 reveal that on the FGVCAircraft dataset, both zero-shot CLIP and linear probe CLIP demonstrate notably low performance, indicating the inadequacy of CLIP's pre-trained features for this dataset. Conversely, the significant improvement with linear probe CLIP on the EuroSAT dataset highlights the generalization potential of CLIP's pre-trained features on this distribution.

In practical prompt learning scenarios, samples from the target dataset distribution are unavailable. Therefore, we evaluate the generalization potential of CLIP's pre-trained features on each dataset by averaging the performance of zero-shot CLIP and few-shot linear probe CLIP.

### D.2 Datasets Representation Similarity Experiment

| | StanfordCars | Flowers102 | FGVCAircraft | SUN397 | DTD | EuroSAT | UCF101 |
|---|---|---|---|---|---|---|---|
| Visual Similarity | 0.3289 | 0.3433 | 0.2781 | 0.4359 | 0.4097 | 0.5103 | 0.2924 |
| Textual Similarity | 0.1253 | 0.1910 | 0.2017 | 0.2407 | 0.2196 | 0.3014 | 0.1993 |
| Average Similarity | 0.2271 | 0.2672 | 0.2399 | 0.3383 | 0.3147 | 0.4059 | 0.2459 |

Table 13: **Estimated similarity between the target dataset and the ImageNet distribution.** The similarity refers to the minimum pairwise cosine similarity between category representations of the target datasets and ImageNet.

**Experimental settings.** Since prompt learning adapts to downstream tasks through few-shot learning using frozen CLIP pre-trained features, we measure the similarity between target datasets and ImageNet using representations from the vision and text encoders of zero-shot CLIP. Specifically, for visual similarity, we calculate the pair-wise cosine similarity between the average representation of test images from each category in the target dataset and the average representation of few-shot images from each category in ImageNet, selecting the minimum value as the measure of visual similarity. For textual similarity, we use the fixed template "a photo of label" as input, compute the pair-wise cosine similarity between text representations of each category in the target dataset and ImageNet, and again select the minimum value as the measure of textual similarity. Ultimately, the average of visual and textual similarities is taken as the estimated similarity between the distributions of the target dataset and ImageNet.

The results in Table 13 demonstrate that the FGVCAircraft dataset exhibits low similarity with ImageNet, aligning with observations from the experiment where prompt-based fine-tuning algorithms generally underperform zero-shot CLIP on FGVCAircraft in cross-dataset generalization settings. Conversely, the EuroSAT dataset shows higher similarity to the ImageNet distribution, which partially explains the differing performance of M$^3$PL on these datasets.

### D.3 Multivariate Linear Regression

|  | StanfordCars | Flowers102 | FGVCAircraft | SUN397 | DTD | EuroSAT | UCF101 |
|---|---|---|---|---|---|---|---|
| Average Performance | 0.7306 | 0.8401 | 0.3549 | 0.6788 | 0.5712 | 0.6737 | 0.7513 |
| Average Similarity | 0.2271 | 0.2672 | 0.2399 | 0.3383 | 0.3147 | 0.4059 | 0.2459 |
| Performance Gain (%) | 0.84 | 1.53 | -0.43 | 5.05 | 1.52 | 9.12 | 0.86 |

Table 14: Two metrics and performance gains of M$^3$PL compared to zero-shot CLIP.

In this section, we use multivariate linear regression to explain the performance improvements of our proposed M$^3$PL model (relative to zero-shot CLIP), based on two metrics derived from the previous sections. The first metric measures the generalization potential of CLIP's original pre-trained features on the target dataset, indicated by the average performance of zero-shot CLIP and few-shot linear probe CLIP. The second metric, the average cosine similarity of textual and visual representations, estimates the distribution similarity between the target dataset and ImageNet. We utilize these two metrics as independent variables in a simple multivariate linear model, with the performance gain of M$^3$PL as the dependent variable.

$$\boldsymbol{Y} = \alpha_0 \cdot \boldsymbol{X}_0 + \alpha_1 \cdot \boldsymbol{X}_1 + \boldsymbol{\beta} \tag{32}$$

where $\alpha_i$ is the regression coefficient and $\boldsymbol{\beta}$ is the intercept, $\boldsymbol{X}_0$ represents the average performance (informativeness), $\boldsymbol{X}_1$ the average similarity (transferability), and $\boldsymbol{Y}$ the performance gain of M$^3$PL.

The fitting results in a Multiple R of 0.952 and an $R^2$ of 0.906, indicating a strong fit and demonstrating the interpretability of our method regarding the performance on the target dataset. Furthermore, both coefficients $\alpha_0 = 3.647$ and $\alpha_1 = 47.429$ are positive, suggesting that the performance improvement of M$^3$PL on a given target dataset positively correlates with both the generalization potential of CLIP's pre-trained features on that dataset and the dataset's similarity to ImageNet.

As shown in Table 32, the EuroSAT and SUN397 datasets exhibit high average performance and average similarity metrics, which correlate with their significant performance enhancements. Conversely, the FGVCAircraft dataset shows lower values in these metrics, resulting in the poorest performance of M$^3$PL. The StanfordCars and UCF101 datasets, while having high average performance, are constrained by low average similarity, limiting their gains to less than 1%. In contrast, the DTD and Flowers datasets benefit from higher average similarity and average performance, respectively, achieving improvements exceeding 1.5%.

