# OpenReview forum: "M$^3$PL: Identifying and Exploiting View Bias of Prompt Learning"
_TMLR — Accepted by TMLR_

### Review · Reviewer_bZhj · 2024-06-08

**Summary Of Contributions:**

This paper first poses an interesting phenomenon that existing prompt learning methods fail to generalize well under large distribution shifts. It then provides a theoretical analysis of multi-modal attention-based prompt learning and highlights the view bias in prompt learning. Finally, the paper proposes a new prompt-learning pipeline that introduces multiple, paired multimodal prompts and achieves state-of-the-art performance.

**Audience:**

Yes

**Broader Impact Concerns:**

Broader impact concerns are missing; please provide a discussion on it.

**Claims And Evidence:**

Yes

**Requested Changes:**

- It would be beneficial to see the efficiency of the proposed method, e.g., adding running time in Table 2.
- Include a discussion on the difficulty of addressing (removing) the assumptions in the theoretical part.
- Provide a brief introduction of each dataset used in the appendix.
- Provide more details on the experimental setup, e.g., the hyperparameters set up in prompt learning, and how to create different prompts.
- Minor request: Is it possible to show a similar phenomenon on large vision language model? ( it would be great to see the result or just  a brief discussion on them.)
- Provide a discussion on the broader impact of the paper.

**Strengths And Weaknesses:**

Strengths:
-  The paper is well-organized and centered on the OOD problem of prompt learning, equipped with both theoretical analysis and strong empirical validation.
- The theoretical analysis in Sections 5.1-5.2 is interesting, extending prompt learning from single-modal [1] to multi-modal.
- The proposed methods are well-motivated and achieve strong performance.
- The experimental results are comprehensive, with a number of datasets and baselines.
- The code is provided in the supplementary Material.

Weaknesses:
- The proposed method might be time-consuming as it requires learning lots of different prompts in numerous layers.
- Proposition 1 is a trivial result in my view. It is straightforward to see that the objective can be decomposed into these four terms: text prompt-vision prompt, vision representation-vision representation, text prompt-vision representation, and vision prompt-text representation. However, the so-called feature selection is too broad as it does not clarify how these weights select the features. Why do different prompts select different features?
- The considered multimodal model is only CLIP-based classifier and does not cover vision language models.
- The theoretical analysis should cover a more practical setting. For example, 1) inner-product is considered as sim() while in practice cosine similarity is used; the difficulty and possible ways to extend this should be discussed. 2) The implications of the assumptions in Section 5.2 should be remarked (do they hold in practice?)

---

> ### Public Comment · ~Yoontae_Hwang1 · 2024-07-17
> **Reproducibility**
>
> In your review you said that a code is provided, but this paper doesn't seem to have one.

---

> > ### Author Response · Authors · 2024-07-25
> > **Response to Public Comment**
> >
> > Our code is provided in the supplementary materials, which will not be made public until the review process is completed.

---

> ### Author Response · Authors · 2024-07-25
> **Response to Reviewer bZhj**
>
> **W1:** The proposed method might be time-consuming as it requires learning lots of different prompts in numerous layers.
>
> **A1:** **(1) Prompt depth:** When prompts are inserted only at the input layer, their corresponding output tokens still participate in subsequent block computations. Additionally, the extra learnable parameters introduced by inserting prompts in deeper layers are negligible compared to the total parameters of CLIP. As mentioned in [1], the baseline method IVLP, which inserts prompts in more layers, and CoOp, which inserts prompts only at the input layer, have nearly identical floating-point operations (FLOPs). Therefore, inserting prompts in more layers does not significantly increase computational overhead compared to inserting prompts only at the input layer. **(2) The number of prompts:** Learning more prompts does lead to a higher computational cost. Consequently, we reduced the number of training iterations to ensure that the training time of M3PL is comparable to that of the previous state-of-the-art method, PromptSRC. Following [2], we measure the practical training time and inference speed of different prompt learning methods on the SUN397 dataset in the base-to-new generalization setting. The results are shown in the table below.
>
> |  | CoOp | CoCoOp | MaPLe | PromptSRC | M$^3$PL (Ours, $M=8$)|
> |:-- | :--: | :--:  |  :--:  |   :--:  |   :--:  |
> |Training time (min)| 8.52  | 56.23  | 6.17  | 22.85     | 21.77|
> |Inference time (images/s) | 323.1 | 19.0     | 328.2 | 329.7     | 183.4|
>
> We greatly appreciate your constructive feedback. In response, we have added a computational analysis part in Section 6.6 of the main text. You can check this section in the revised manuscript for more details and related discussions.
>
> **W2:** Proposition 1 is a trivial result in my view... Why do different prompts select different features?
>
> **A2:** As we stated in Section 5.2, the main reason for different prompts selecting different features is the multi-solution nature of the representation matching objective. By decomposing the objective, Proposition 1 actually makes this point more clear---if we assume that the visual representation $\boldsymbol{v}$ is linearly composed of a set of features $\{\boldsymbol{f}_1,\ldots,\boldsymbol{f}_l\}$, then matching the textual prompt $\boldsymbol{p}_t$ and $\boldsymbol{v}$ by pushing $\boldsymbol{W}_t^\top\boldsymbol{p}_t$ along the direction of any label-relevant feature $\boldsymbol{f}_i,i\in\{1,\ldots,l\}$ can reduce the loss. Different prompts thus not necessarily select the same same feature subset. Please refer to **A4** for more discussions on the assumptions.
>
> **W3:** The considered multimodal model is only CLIP-based classifier and does not cover vision language models.
>
> **A3:** Yes, our current experiments are limited to visual recognition tasks based on the CLIP model. However, our theoretical analysis and methods have the potential to generalize to a broader range of tasks and model scales. We will leave the exploration of these directions for future work and have added a relevant discussion in Section 7 of the revised manuscript.
>
> **W4:** The theoretical analysis should cover a more practical setting. For example, 1) inner-product is considered as sim() while in practice cosine similarity is used; the difficulty and possible ways to extend this should be discussed. 2) The implications of the assumptions in Section 5.2 should be remarked (do they hold in practice?)
>
> **A4:** Thank you for your constructive suggestions. We address your concerns below and have modified the manuscript accordingly.
>
> 1) We note that inner-product and cosine similarity are *equivalent* if we normalize the representations before calculating the loss. Prompt learning using cosine similarity can thus be viewed as feature selection on _normalized features_. In practice, normalizing the representations often results in comparable classification performance to using unnormalized representations (e.g., see a relevant dicussion from the offical repository of CLIP: https://github.com/openai/CLIP/issues/85).
>
> 2) The main assumptions in Section 5.2 are (1) a linear combination of different features in the representations and (2) gradient descent tending to extract only a random subset of features. The first assumption is common in analyzing the feature learning process of neural networks, and prior work has shown that it can capture many practical feature learning characteristics (see e.g., [3,4,5]). The second assumption is related to the well-known simplicity bias of neural networks (see e.g., [6]); while we provide empirical evidence (Section 4) suggesting that this assumption also holds in our setup, rigorously proving it is challenging due to the requirement of analyzing fine-grained gradient descent dynamics, which we think may warrant another paper. We have added those discussions to the revised manuscript.

---

> > ### Author Response · Authors · 2024-07-25
> > **Response to Reviewer bZhj**
> >
> > **R1:** It would be beneficial to see the efficiency of the proposed method, e.g., adding running time in Table 2.
> >
> > **A5:** Thank you for your constructive suggestion. We have incorporated an analysis of computational overhead in Section 6.6 of the main text. Please refer to **A1** and the revised manuscript for details on the changes.
> >
> > **R2:** Include a discussion on the difficulty of addressing (removing) the assumptions in the theoretical part.
> >
> > **A6:** We have added a discussion on extending the $\mathrm{sim}()$ function to cosine similarity in Section 5.1 (after Eq.(6), highlighed in blue) and a discussion on proving the assumptions stated in Section 5.2 (before Proposition 2, highlighed in blue).
> >
> > **R3:** Provide a brief introduction of each dataset used in the appendix.
> >
> > **A7:** Thank you for your suggestion. We have added a brief introduction of all the datasets used in our study in section C.2 of the appendix in the revised manuscript.
> >
> > **R4:** Provide more details on the experimental setup, e.g., the hyperparameters set up in prompt learning, and how to create different prompts.
> >
> > **A8:** Thank you for your suggestion. For M$^3$PL, we use a normal distribution with a mean of zero to randomly initialize the prompts, and increase the variance with the number of prompts ($M$) to ensure diversity. In Appendix C.1, we detail the specific hyperparameter settings for prompt learning under different generalization scenarios. Additionally, the configuration files used in our experiments are included in the released code to ensure the reproducibility of our results.
> >
> > **R5:** Minor request: Is it possible to show a similar phenomenon on large vision language model? ( it would be great to see the result or just a brief discussion on them.)
> >
> > **A9:** While our empirical investigations are restricted to CLIP, we expect similar findings may also be obtained in larger models. In particular, a large number of existing VLMs such as LLaVa, MiniGPT-4 and InstructBLIP directly leverage CLIP and its derivatives as image encoders. It has been shown that fine-tuning those image encoders can lead to worse OOD performance [7], similar to our empirical observation that the OOD accuracy can drop after prompt learning. We thus envision that our results may be extended to those VLMs, but leave rigorous investigations as future work due to computation constraints. We have also included the above discussion in Section 4 in the revised paper.
> >
> > **R6:** Provide a discussion on the broader impact of the paper.
> >
> > **A10:** We have included an additional "Broader Impact Statement" section in the main text (before references) in the revised paper.
> >
> >
> > ### References
> >
> > [1] MaPLe: Multi-Modal Prompt Learning. Khattak et al., CVPR, 2023.
> >
> > [2] Self-regulating prompts: Foundational model adaptation without forgetting. Khattak et al., ICCV, 2023.
> >
> > [3] Towards understanding ensemble, knowledge distillation and self-distillation in deep learning. Allen-Zhu et al., ICLR, 2023.
> >
> > [4] Provable multi-task representation learning by two-layer ReLU neural networks. Collins et al., ICML, 2024.
> >
> > [5] Feature contamination: Neural networks learn uncorrelated features and fail to generalize. Zhang et al., ICML, 2024.
> >
> > [6] The pitfalls of simplicity bias in neural networks. Shah et al., NeurIPS, 2020.
> >
> > [7] Prismatic VLMs: Investigating the design space of visually-conditioned language models. Karamcheti et al., ICML, 2024.

---

> > > ### Comment · Reviewer_bZhj · 2024-07-27
> > > **Thanks for the response**
> > >
> > > I appreciate the detailed clarification from the authors and will reconsider them during the final evaluation of this work.

---

### Review · Reviewer_TUPP · 2024-06-09

**Summary Of Contributions:**

This paper identifies the problem of view bias in multi-modal prompt learning, where the learned prompts represent a partial subset of useful features. The existence of view bias in learned prompts leads to varying behaviors under distribution shifts. The authors showed theoretical analysis into the view bias problem and connect it to the multi-solution nature of prompt learning. Based on these observations, the authors proposed a new method M3PL (Multi-modal matching multi-prompt learning) to encourage learning diverse views of features.

**Audience:**

Yes

**Broader Impact Concerns:**

I do not have concerns regarding the broader impacts of this paper.

**Claims And Evidence:**

Yes

**Requested Changes:**

See weaknesses for details. But there are two specific points that I would prefer to be addressed:

1. Can the authors modify section 4 into a self-contained section? Currently it is very short and I guess the initial evidence refers to the Figure 1 described in the introduction. This seems confusing for paper presentation. It would be good to dedicate this section to formally investigate the view bias problem. You can give an overview of Figure 1 in the introduction but make sure to defer the technical details and discussions in the main text.

2. More discussions on the computational cost should be put in the main text. In the experiment section, the authors should discuss how the number of prompts 8 is chosen. Also it would be good to discuss how does the use of more prompts in M3PL affect the computational efficiency.

**Strengths And Weaknesses:**

Strengths:
1. This paper is well-written and easy to understand. I like that the paper looks into a specific problem in existing prompt learning methods and use the analysis to propose a new method.

2. The authors discover and analyze an intriguing problem of existing multi-modal prompt learning methods, where the learned prompts capture a limited set of features. This phenomenon should be of interest for the research community.

3. The proposed method is intuitive and technically sound. The authors have demonstrated the effectiveness of the proposed method M3PL on many datasets.


Weaknesses:
1. The use of multiple prompts in both the image and text encoder can make the proposed approach computationally heavy. However, the paper does not have discussions on this practical perspective. (It is merely discussed with one line in Section 6.6.)

2. It is not clear to me how to choose the number of prompts M in the proposed methods. I couldn’t find any mentions of this in the main text, e.g., how is M=8 chosen empirically in the experiments?

3. I wonder if the proposed method actually solves or alleviates the view bias problem. Is there a way to test this, similar to the analysis in Figure 1 and Figure 2?

4. The authors only test on one CLIP pretrained model. It would be good to show the results on other variants of CLIP (e.g. SigLIP [1]) to demonstrate the generalization ability of the proposed approach.

5. There are no results on ImageNet correuption datasets (e.g. ImageNet-C, ImageNet-R), which are representative datasets for measuring OOD robustness of CLIP models [2,3].

[1] Sigmoid loss for language image pre-training. Zhai et al, 2023.
[2] Robust fine-tuning of zero-shot models. Wortsman et al, 2021.
[3] Finetune like you pretrain: improved finetuning of zero-shot vision models. Goyal et al, 2023.

---

> ### Author Response · Authors · 2024-07-25
> **Response to Reviewer TUPP**
>
> **W1:** The use of multiple prompts in both the image and text encoder can make the proposed approach computationally heavy. However, the paper does not have discussions on this practical perspective.
>
> **A1:** We follow the MaPLe's [1] setup and set the prompt length of the visual and language branches to half that in uni-modal prompt learning methods such as CoOp. According to [1], the floating-point operations (FLOPs) of the single-prompt multimodal prompt learning baseline method, IVLP, are comparable to those of CoOp. Consequently, the overall FLOPs of M$^3$PL are approximately $M$ (the number of prompts) times those of CoOp. To address the increased time cost associated with learning multiple prompts, we reduce the training iterations, making the total training time of M$^3$PL comparable to the previous SOTA method, PromptSRC. Following [3], we measure the practical training time and inference speed of different prompt learning methods on the SUN397 dataset in the base-to-new generalization setting. The results are shown in the table below.
>
> |  | CoOp | CoCoOp | MaPLe | PromptSRC | M$^3$PL (Ours, $M=8$)|
> |:-- | :--: | :--:  |  :--:  |   :--:  |   :--:  |
> |Training time (min)| 8.52  | 56.23  | 6.17  | 22.85     | 21.77|
> |Inference time (images/s) | 323.1 | 19.0     | 328.2 | 329.7     | 183.4|
>
> We greatly appreciate your constructive suggestion. In response, we have added a computational analysis part in Section 6.6 of the main text. You can check this section in the revised manuscript for more details and related discussions.
>
> **W2:** It is not clear to me how to choose the number of prompts M in the proposed methods. I couldn't find any mentions of this in the main text, e.g., how is $M=8$ chosen empirically in the experiments?
>
> **A2:** Based on the ID and OOD performance with different numbers of prompts on the held-out validation set, we set $M$ to 8 in the experiments to achieve optimal OOD accuracy while maintaining satisfactory ID accuracy. In Section 6.6 of the revised main text, we have added an ablation study on the number of prompts and provided the test performance of M$^3$PL for different choices of $M$ in Figure 4. You can refer to the corresponding sections of the revised manuscript for more detailed settings and related discussions.
>
> **W3:** I wonder if the proposed method actually solves or alleviates the view bias problem. Is there a way to test this, similar to the analysis in Figure 1 and Figure 2?
>
> **A3:** Following the analysis in Figure 1, we extend the experiments in Table 1. Specifically, we calculate the relative average Jensen-Shannon (JS) divergence between the predicted label distribution of different aggregated prompts trained with M3PL for OOD misclassified samples, to verify whether our proposed M3PL method effectively alleviates the view bias problem. The results are presented in the table below.
>
> ||Caltech101|OxfordPets|StanfordCars|Flowers102|Food101|FGVCAircraft|SUN397|DTD|EuroSAT|UCF101|Average|
> |:--:|:--:|:--:|:--:|:--:|:--:|:--:|:--:|:--:|:--:|:--:|:--:|
> |Average JS Divergence|1.043|0.520|0.590|0.850|0.254|1.470|0.358|0.516|1.000|0.579|0.718|
> |+M$^3$PL(Ours)| 0.364 | 0.091 | 0.079 | 0.155 | 0.031 | 0.303 | 0.055 | 0.083 | 0.117 | 0.144 | 0.142 |
> |$\Delta$ | -65\% | -83\% | -87\% | -82\% | -88\% | -79\% | -85\% | -84\% | -88\% |-75\% | -80\% |
>
> We observe that the JS divergence significantly decreases
> across all datasets, with an average reduction of 80%, clearly demonstrating the effectiveness of our method
> in mitigating the view bias problem. You can check Section 4 in the revised manuscript for more details and related discussions.
>
> **W4:** The authors only test on one CLIP pretrained model. It would be good to show the results on other variants of CLIP (e.g. SigLIP [2]) to demonstrate the generalization ability of the proposed approach.
>
> **A4:** Theoretically, our analysis can be extended to the sigmoid loss used by SigLIP [2]. To see this, note that the crux of our analysis is exploiting the inner-product term between multi-modal representations to show the multi-solution property of the objective. Recall that the sigmoid loss takes the form of $\log\frac{1}{1 + e^{y(-t\langle\tilde{\boldsymbol{v}},\tilde{\boldsymbol{t}}\rangle + b)}}$, where $y$ denotes whether the visual representation $\tilde{\boldsymbol{v}}$ and textual representation $\tilde{\boldsymbol{t}}$ are matched, $t$ is the temporature, and $b$ is a learnable bias. Our main argument can thus be similarly applied since this loss also has an inner-produce term $\langle\tilde{\boldsymbol{v}},\tilde{\boldsymbol{t}}\rangle$. Empirically, we are unable to directly apply our proposed M$^3$PL method to SigLIP since the official SigLIP checkpoint (from Huggingface) does not yet support fine-tuning [6].
> We thus will leave the verification of our method's generalizability on other VLMs to future work.

---

> ### Author Response · Authors · 2024-07-25
> **Response to Reviewer TUPP**
>
> **W5:** There are no results on ImageNet correuption datasets (e.g. ImageNet-C, ImageNet-R), which are representative datasets for measuring OOD robustness of CLIP models [4,5].
>
> **A5:** Thank you for your suggestion, we fully agree with this point. Therefore, following [4,5], we few-shot fine-tune CLIP on ImageNet and test its zero-shot performance on a set of ImageNet corruption datasets to validate the effectiveness of our approach. The results are presented in the table below.
>
> ||ImageNet (ID)|ImageNetV2|ImageNet-Sketch|ImageNet-A|ImageNet-R|Average|$\rho(f)$|
> |:--|:--:|:--:|:--:|:--:|:--:|:--:|:--:|
> |Zero-shot CLIP|66.68|60.91|46.09|47.76|73.97|57.18|-|
> |IVLP|**71.16**|63.69|48.41|50.57|76.78|59.86|60%|
> |M$^3$PL(Ours)|70.95|**64.49**|**49.60**|**51.47**|**77.40**|**60.74**|**83%**|
>
> Our proposed M$^3$PL method also performs effectively on the ImageNet corruption datasets, showing significant improvements over the zero-shot CLIP and the multimodal prompt learning baseline, IVLP. We categorize these experiments under domain generalization, and more details about the experimental setup and comparisons with other prompt learning methods can be found in Table 8 of Appendix C.3.3 in the revised manuscript.
>
> **R1:** Can the authors modify section 4 into a self-contained section? Currently it is very short and I guess the initial evidence refers to the Figure 1 described in the introduction. This seems confusing for paper presentation. It would be good to dedicate this section to formally investigate the view bias problem. You can give an overview of Figure 1 in the introduction but make sure to defer the technical details and discussions in the main text.
>
> **A6:** Thank you for your constructive suggestion. We have expanded and rearranged Section 4 to include more technical details and discussions on our empirical investigations regarding the view bias problem.
>
> **R2:** More discussions on the computational cost should be put in the main text. In the experiment section, the authors should discuss how the number of prompts 8 is chosen. Also it would be good to discuss how does the use of more prompts in M3PL affect the computational efficiency.
>
> **A7:** We have added discussions on computational costs and the number of prompts in Section 6.6 of the main text. Details of these modifications can be found in **A1** and **A2**, as well as in the revised manuscript.
>
> ### References
>
> [1] MaPLe: Multi-Modal Prompt Learning. Khattak et al., CVPR, 2023.
>
> [2] Sigmoid loss for language image pre-training. Zhai et al., ICCV, 2023.
>
> [3] Self-regulating prompts: Foundational model adaptation without forgetting. Khattak et al., ICCV, 2023.
>
> [4] Robust fine-tuning of zero-shot models. Wortsman et al., CVPR 2022.
>
> [5] Finetune like you pretrain: improved finetuning of zero-shot vision models. Goyal et al., CVPR, 2023.
>
> [6] https://huggingface.co/docs/transformers/model_doc/siglip

---

### Review · Reviewer_dPz4 · 2024-07-17

**Summary Of Contributions:**

In this paper, the authors propose Multi-modal Matching Multi-Prompt Learning (M3PL) to improve the generalization ability of prompt tuning. The key idea is to leverage multiple paired prompts and a cross-modal contrastive regularization that facilitates the prompt pairs to comprehensively utilize the features. The effectiveness is verified on several standard prompt tuning datasets.

**Audience:**

Yes

**Claims And Evidence:**

Yes

**Requested Changes:**

Please check weakness for details.

**Strengths And Weaknesses:**

Pros:
1. The identified problem is interesting.
2. The proposed solution is straight-forward.
3. The result is convincing.

Cons:
1. The proposed method achieves a much more significant improvement on EuroSat compared to on other datasets. This raise a natural question: does the substantial improvement really comes from the proposed theory?

a. It is necessary to further examine other datasets with rather large domain shift.

b. It is important to further examine the EuroSat dataset and the resulted model on it. Is it possible that this dataset has some hidden shortcut? For example, the dataset has a spatial prior between image regions and classes?

2. The current experiment is only done with one backbone. It is necessary to check the behavior when scaling up to larger and better backbones.

3. The proposed method is a generalized form of [a]. But the effect on using multiple text prompts is different from [a]. It is worth examination and discussion for such discrepancy.

[a] Learning to Decompose Visual Features with Latent Textual Prompts

---

> ### Author Response · Authors · 2024-07-25
> **Response to Reviewer dPz4**
>
> **W1:** The proposed method achieves a much more significant improvement on EuroSat compared to on other datasets. This raise a natural question: does the substantial improvement really comes from the proposed theory?
>
> **W1(a):** It is necessary to further examine other datasets with rather large domain shift.
>
> **W1(b):** It is important to further examine the EuroSAT dataset and the resulted model on it. Is it possible that this dataset has some hidden shortcut? For example, the dataset has a spatial prior between image regions and classes?
>
> **A1:** Indeed, we have observed this more significant improvement of our M$^3$PL model on the EuroSAT dataset under the cross-dataset generalization setting as you mentioned. To elucidate the relationship between this experimental phenomenon and our theory, we have conducted further analysis in Section 6.5 of the main text and Appendix D. In brief, according to our Proposition 1, optimizing the representation matching objective can be seen as a feature selection process. Thus, the improvement of M3PL on specific datasets depends not only on the prompt learning process but also on the quality and transferability of the pre-trained CLIP features and the learnable features on ID data (few-shot ImageNet data in our settings). We design two metrics to measure these factors, as shown in Table 13. The EuroSAT dataset scores highly on both metrics, leading to the most significant improvement. Please refer to the relevant sections for more details.
>
> **A1(a):**
> As mentioned above, we have conducted experimental analysis to explain the varying improvements of our method in different datasets in the cross-dataset generalization setting. Moreover, we note that in the base-to-new generalization setting, our method achieves consistent and significant improvements on multiple datasets with large distribution shifts such as FGVC Aircraft, DTD, EuroSAT, and UCF-101.
>
> **A1(b):**
> We would like to note that unlike [1] and CoOp [2], in the cross-dataset setting we fine-tune our model on the ImageNet dataset and then perform **zero-shot** testing directly on the target dataset (instead of using few-shot samples from the target dataset for fine-tuning). Therefore, the model cannot leverage shortcuts such as spatial priors that are specific to the test set during fine-tuning.
>
> **W2:** The current experiment is only done with one backbone. It is necessary to check the behavior when scaling up to larger and better backbones.
>
> **A2:** We follow the convention in existing prompt learning works [2, 3, 4] to use CLIP ViT-B/16 as the main backbone. Although some work such as [1] also considers other backbones such as CLIP RN-50 and CLIP ViT-B/32, those backbones are smaller and thus do not perform as well as our backbone. Due to computational constraints, it is hard for us to further scale up the backbone, yet we believe that the scale of our experiments already meets the common standard in the prompt learning literature.
>
> **W3:** The proposed method is a generalized form of [1]. But the effect on using multiple text prompts is different from [1]. It is worth examination and discussion for such discrepancy.
>
> **A3:** We appreciate your insightful comment, as it allows us to clarify the contributions of our work further. The primary distinctions between our results and [1] are: (1) Experimental setup: [1] conducts few-shot learning directly on multiple target datasets. In contrast, our cross-dataset generalization setup involves fine-tuning the model on ImageNet in a few-shot manner, followed by zero-shot testing on the target datasets, emphasizing the model’s out-of-distribution (OOD) generalization capability. (2) Prompt learning paradigm: our approach adheres to the CoOp paradigm by fixing the class label and optimizing a learnable prefix, whereas [1] treats the class label as a learnable token and subsequently trains an additional linear classification head on the ID data. Therefore, our conclusions are not contradictory. Future work could potentially integrate both methodologies. We have incorporated this discussion into the related work section of the revised manuscript.
>
> ### References
>
> [1] Learning to Decompose Visual Features with Latent Textual Prompts. Wang et al., ICLR, 2023.
>
> [2] Learning to Prompt for Vision-Language Models. Zhou et al., IJCV, 2022.
>
> [3] MaPLe: Multi-Modal Prompt Learning. Khattak et al., CVPR, 2023.
>
> [4] Self-regulating prompts: Foundational model adaptation without forgetting. Khattak et al., ICCV, 2023.

---

### Review · Reviewer_k5Sg · 2024-07-23

**Summary Of Contributions:**

This paper examines the limitations of multi-modal prompt learning, particularly the tendency of prompts to capture only a subset of features, termed "view bias." This bias often undermines model generalization, especially under distribution shifts. To address this, the authors introduce Multi-modal Matching Multi-Prompt Learning (M3PL), a new framework that uses multiple prompt pairs and a cross-modal contrastive regularizer, thereby enhancing model generalization when facing OOD data. Extensive experiments demonstrate that M3PL significantly improves OOD performance, offering a robust solution to the identified limitations of existing prompt learning techniques in vision-language models

**Audience:**

Yes

**Broader Impact Concerns:**

n.a.

**Claims And Evidence:**

Yes

**Requested Changes:**

Please make changes in accord with aforementioned weaknesses.

**Strengths And Weaknesses:**

**Strengths:**

1. This paper studies an interesting yet often overlooked problem; the authors describe this as the "view bias", which manifests that a single fixed prompt can only help the model select part of the features.

2. The theoretical analysis seems correct, the proof and the results seem novel to me, and the implications from Proposition 2 and Proposition 3 seem interesting and non-trivial.

3. The proposed method showcases significant improvements over OOD scenarios.

**Weaknesses:**

1. Proposition 2 is somewhat hard to grasp; based on my understanding, the definition of "generalization" for general-purpose pre-trained models such as CLIP is not as clearly defined as a supervised learning setup. I recommend authors to

2. The problem setup seems to differ from my understanding of prompt tuning in CLIP, based on the original setup proposed in Coop [1]; only the text branch prompt is learnable. Do authors adapt to a more trending setting?

3. What approaches are undertaken to make sure that different prompt branches are trained *independently*? This does not seem feasible unless (a) those learnable prompts are defined over different hypothesis classes and (b) those prompts are trained with different data.

[1] Learning to Prompt for Vision-Language Models

---

> ### Author Response · Authors · 2024-07-25
> **Response to Reviewer k5Sg**
>
> **W1:** Proposition 2 is somewhat hard to grasp; based on my understanding, the definition of "generalization" for general-purpose pre-trained models such as CLIP is not as clearly defined as a supervised learning setup. I recommend authors to
>
> **A1:** Unfortunately the last sentence in the review is not complete, but we assume that you meant to suggest we clarify the definition of generalization in our setup. Briefly speaking, here we adopt a similar definition as in standard supervised learning: we fine-tune the pre-trained model using a training set (drawn from a training distribution) and measure the generalization performance of the fine-tuned model on a test distribution that is different from the training distribution (rather than the _pre-training_ distribution).
>
>
> **W2:** The problem setup seems to differ from my understanding of prompt tuning in CLIP, based on the original setup proposed in CoOp [1]; only the text branch prompt is learnable. Do authors adapt to a more trending setting?
>
> **A2:** Yes, we primarily followed the multimodal prompt learning framework initially proposed by MaPLe [2] and IVLP [3]. The computational cost of this framework is almost the same as adding prompts only to the text branch (for more details, please refer to our response **A1** to Reviewer TUPP), while significantly enhancing the matching flexibility between two modality representations. This advantage inspires our theoretical analysis and algorithm design, leading to our adoption of this approach.
>
> **W3:** What approaches are undertaken to make sure that different prompt branches are trained independently? This does not seem feasible unless (a) those learnable prompts are defined over different hypothesis classes and (b) those prompts are trained with different data.
>
> **A3:** By the term "independent", we meant that the training processes of different prompts do not impact each other. In this way, even if prompts are trained on the same set of data, their parameters after training (as random variables) are still independent. This can be realized by training different prompts in a paralleled manner so that their gradients are computed separately.
>
> ### References
>
> [1] Learning to Prompt for Vision-Language Models. Zhou et al., IJCV, 2022.
>
> [2] MaPLe: Multi-Modal Prompt Learning. Khattak et al., CVPR, 2023.
>
> [3] Fine-tuned clip models are efficient video learners. Rasheed et al., CVPR, 2023.

---

### Author Response · Authors · 2024-07-25
**General Response**

We are grateful for the insightful and constructive feedback from all reviewers. We are glad that all reviewers seem to be positive with our submission and found our work identifying an interesting problem (k5Sg, dPz4, TUPP), our theoretical results novel (k5Sg), our analysis interesting (bZhj), our proposed method well-motivated (bZhj) and technically sound (TUPP), and our results comprehensive (bZhj), significant (k5Sg), and convincing (dPz4). We have provided point-by-point response to all reviewers separately and have accordingly revised our manuscript. The modifications are highlighted in blue in the updated manuscript.

---

### Decision · Action_Editor_JNU6 · 2024-08-17

**Recommendation:** Accept with minor revision

**Comment:**

Even though the authors have explicitly mentioned the limitation "our experiments are currently limited to CLIP ViT-B/16", I would encourage them to consider extending their method beyond this CLIP ViT-B/16 model. If this is not possible, the paper seems ready to be accepted, but I do think the paper would become stronger with an experiment extending beyond this mdoel.

**Audience:**

Yes, the paper is on the timely topic of prompt learning with large multi-modal models, so it will likely be relevant for the TMLR community.

**Claims And Evidence:**

The paper focus on multi-modal prompt learning in CLIP model. The paper reports that only a subset of useful features are extracted from different learned prompts, which might result in problems under distribution shift. The paper introduces Multi-modal Matching Multi-Prompt Learning (MPL) to pair multiple prompts with different features.

The claims of this paper are accurate - the paper demonstrates this bias observed in the CLIP model and then a method to mitigate that is suggested. The evidence for CLIP is provided. Some of the reviewers raise the issue that this paper is too focused on CLIP, and the evidence is supported mainly on the EuroSat dataset. The rebuttal clarifies that scaling to other backbones might be trickier due to computational reasons, which is reasonable to me.